# Non-invasive hydrodynamic imaging in plant roots at cellular resolution

Flavius C. Pascut [1,9✉], Valentin Couvreur [2,9✉], Daniela Dietrich [3,9], Nicky Leftley[4], Guilhem Reyt [4,5], Yann Boursiac [6], Monica Calvo-Polanco [6,8], Ilda Casimiro[7], Christophe Maurel [6], David E. Salt [4,5], Xavier Draye [2], Darren M. Wells [4,5], Malcolm J. Bennett [4,5,10] & Kevin F. Webb [1,10]

A key impediment to studying water-related mechanisms in plants is the inability to non-invasively image water fluxes in cells at high temporal and spatial resolution. Here, we report that Raman microspectroscopy, complemented by hydrodynamic modelling, can achieve this goal - monitoring hydrodynamics within living root tissues at cell- and sub-second-scale resolutions. Raman imaging of water-transporting xylem vessels in *Arabidopsis thaliana* mutant roots reveals faster xylem water transport in endodermal diffusion barrier mutants. Furthermore, transverse line scans across the root suggest water transported via the root xylem does not re-enter outer root tissues nor the surrounding soil when *en-route* to shoot tissues if endodermal diffusion barriers are intact, thereby separating 'two water worlds'.

[1] Optics & Photonics Research Group, Faculty of Engineering, University of Nottingham, Nottingham, UK. [2] Earth and Life Institute, Université catholique de Louvain, Louvain-la-Neuve, Belgium. [3] School of Biological Sciences, University of Bristol, Bristol, UK. [4] Division of Plant and Crop Sciences, School of Biosciences, University of Nottingham, Sutton Bonington, UK. [5] Future Food Beacon of Excellence, University of Nottingham, Sutton Bonington, UK. [6] BPMP, Univ Montpellier, CNRS, INRAE, Institut Agro, Montpellier, France. [7] Departamento de Anatomía, Biología Celular y Zoología, Universidad de Extremadura, Facultad de Ciencias, Badajoz, Spain. [8] Present address: Excellence Unit AGRIENVIRONMENT, CIALE, University of Salamanca, Salamanca, Spain. [9] These authors contributed equally: Flavius C. Pascut, Valentin Couvreur, Daniela Dietrich [10] These authors jointly supervised this work: Malcolm J. Bennett, Kevin F. Webb. ✉email: flavius.pascut@nottingham.ac.uk; valentin.couvreur@uclouvain.be

Water plays an essential role as a solvent for nutrients, minerals, and other biomolecules in plant tissues[1,2]. To date, the inability to non-invasively image and quantify water transport directly within root tissues has been a key stumbling block for researchers seeking to understand hydrodynamics within living plant cells and tissues. Current techniques developed to monitor water uptake in roots either suffer from being indirect (tracking radiotracers[3] or monitoring pressure[4]) or invasive (e.g., pressure chamber[5], root and xylem pressure probes[6], and heat pulsing[7]). The most promising techniques are nuclear magnetic resonance flow imaging[8,9] and rapid neutron tomography[10], which allow non-invasive measurement of structure and water flow within tissues. However, these techniques lack the spatial and temporal resolution to monitor water uptake kinetics at the cellular scale in root tissues.

Raman microspectroscopy (RMS) represents a non-invasive, non-destructive spectroscopic technique with the ability to investigate biological samples in situ, under physiological conditions[11]. By detecting inelastically scattered photons from a narrow-band laser source, RMS provides spectroscopic information at the molecular level without the need for sample preparation, fluorescent labels, or fixation. Biomolecular components (e.g., water, lipids, proteins, nucleic acids, and metabolites) appear as Raman lines at different positions inside the spectrum corresponding to their vibrational energy levels. Since the amount of energy lost by the Raman-scattered photons depends on both mass and geometry of the molecular bonds in the sampling volume, the presence of isotopes such as deuterium can be followed by monitoring distinct Raman shifts.

In this work, vibrational contrast is achieved by monitoring the appearance of deuterated water ($D_2O$) within the Raman "silent region" (void of functional groups between 1800 and 2800 $cm^{-1}$, Supplementary Fig. 1, except for triple-bond vibrations). The spatio-temporal dynamics of water flow is retrieved through an approach combining non-invasive RMS imaging following application of $D_2O$ to the root tip, and inverse modeling of $D_2O$ advection-diffusion down to the cellular scale. Our study explores whether this approach can provide non-invasive measurements of water fluxes in living plant tissues at a cellular (2 µm step size) spatial resolution and sub-second (0.3 s) temporal resolution.

## Results and discussion

### Water en-route to the shoot does not re-enter outer root tissues.
To demonstrate the feasibility of using RMS to non-invasively image the spatial and temporal dynamics of water isotopes in plant tissues, we initially performed a proof of concept study in root tissues of *Arabidopsis thaliana* (Fig. 1 and Supplementary Fig. 2). RMS measurements performed as a line scan in the mature root (Fig. 1A, B, D), 11 mm from the tip, detected a pulse of deuterated water within 80 s of exposing the seedling root tip to a source of $D_2O$ (phase termed "$D_2O$ wash-in"; Fig. 1C, F). Similar temporal dynamics were detected when replacing $D_2O$ with $H_2O$ (phase termed "$D_2O$ wash-out"; Fig. 1E, G). Spatially, the $D_2O$ pulse was detected solely in inner root tissues (i.e., the endodermis and the stele, which contains water-transporting xylem vessels) and not in outer root tissues, at any point of a 64-min experiment composed of wash-in and wash-out phases (Fig. 1C, E). Hence, our results demonstrate that RMS is capable of monitoring hydrodynamics within living plant tissues. Furthermore, this result suggests that water taken up by root tips does not diffuse toward outer layers of mature root tissues in measurable quantities. This is likely to reflect the impact of both the inward advection of $H_2O$ and the diffusion barriers consisting of (i) the lignin belt—termed the Casparian strip (CS)—encircling

endodermal cells within radial cell walls, and (ii) suberin deposition between the endodermal plasma membrane and the cell wall[12,13] (details in Supplementary Table 1). It has been argued that the CS is porous to water since the lignin pore diameter is ~1 nm[14], which is 10 times larger than a water molecule. Hence, the CS acts as a barrier to most solutes but would leave the door open to a fraction of purely apoplastic radial diffusion. Our RMS results do not support water diffusion from the stele to the cortex as *A. thaliana* roots take up water from their environment.

### Quantified xylem water fluxes concur with stomatal aperture data.
Our RMS imaging system offers the opportunity to visualize in situ xylem $D_2O$ dynamics in living roots at unique spatial and temporal resolutions, and to translate these measurements into quantitative xylem water velocities using inverse modeling. In a complementary experiment, in order to parameterize this model at an improved temporal resolution, the laser was directed at a single protoxylem vessel (identified as a local maximum of $D_2O$ signal between wash-in and wash-out phases), and $D_2O$ hydrodynamics was captured at 1 Hz from a diffraction-limited volume.

RMS imaging was used to investigate alterations of $D_2O$ dynamics and xylem water velocity in *A. thaliana* genotypes impaired in the integrity of their endodermal diffusion barriers. Measurements were performed on roots of wild-type plants (WT, Col-0); a WT line expressing the *pCASP1::CDEF* construct which degrades endodermal suberin (termed *CDEF*)[15]; and the *sgn3-3 myb36-2* mutant, which displays no CS but normal suberization[16]. RMS imaging of "$D_2O$ wash-out" from individual protoxylem vessels revealed contrasting hydrodynamic behaviors. For example, the *CDEF* line needed the shortest time to complete $D_2O$ wash-out (linear slope of the wash-out curve: $-5.2 \times 10^{-3} \pm 0.7 \times 10^{-3} s^{-1}$, $N = 18$, significantly steeper than in WT: $-3.9 \ 10^{-3} \pm 0.9 \ 10^{-3} s^{-1}$, $N = 24$, Anova-1 *p*-value $< 10^{-4}$), whereas *sgn3-3 myb36-2* had almost no effect on the wash-out slope ($-4.4 \times 10^{-3} \pm 1.1 \times 10^{-3} s^{-1}$, $N = 9$) compared to WT (Fig. 2F, where traces are normalized to start at a unit value).

From the physical point of view, xylem $D_2O$ wash-out dynamics is determined by water flow (i.e., advection, both axially and laterally) and by the mixing of $H_2O$ and $D_2O$ as they diffuse across the root composite pathways (Fig. 2A, D). Translating our wash-out traces into xylem water flow rates, therefore, required a framework capturing these hydraulic processes. We exploited a recently developed model of the root "hydraulic anatomy" termed MECHA[17]. This model includes explicit tissue geometries, subcellular hydraulic properties (cell walls, membranes, and plasmodesmata), and localization of diffusion barriers (Fig. 2A, B). In the Methods section, we present the implementation of its three-dimensional $D_2O$ advection-diffusion equations. MECHA can simulate $D_2O$ spatiotemporal dynamics during wash-in and wash-out cycles (Fig. 2C, D) in WT, *CDEF*, and *sgn3-3 myb36-2* hydraulic anatomies under physiological conditions (e.g., snapshots at the laser focal point before wash-out starts in Supplementary Fig. 3, showing $D_2O$ leakage in *sgn3-3 myb36-2* due to the absence of the CS). Conducting such simulations first required the estimation of hydraulic conductivity and diffusivity parameters values using an inverse modeling scheme based on an iterative search algorithm (loop indicated by curved solid arrows between panels in Fig. 2) that fine-tunes these values until convergence between measured and simulated variables. The convergence between root hydraulic conductivities (termed $Lp_r$) simulated and measured with a pressure chamber[5] under control and azide treatments (inactivating aquaporins[18]) (Fig. 2E, see individual points in Supplementary Fig. 4) drove the search algorithm to optimal values for the three subcellular hydraulic conductivity

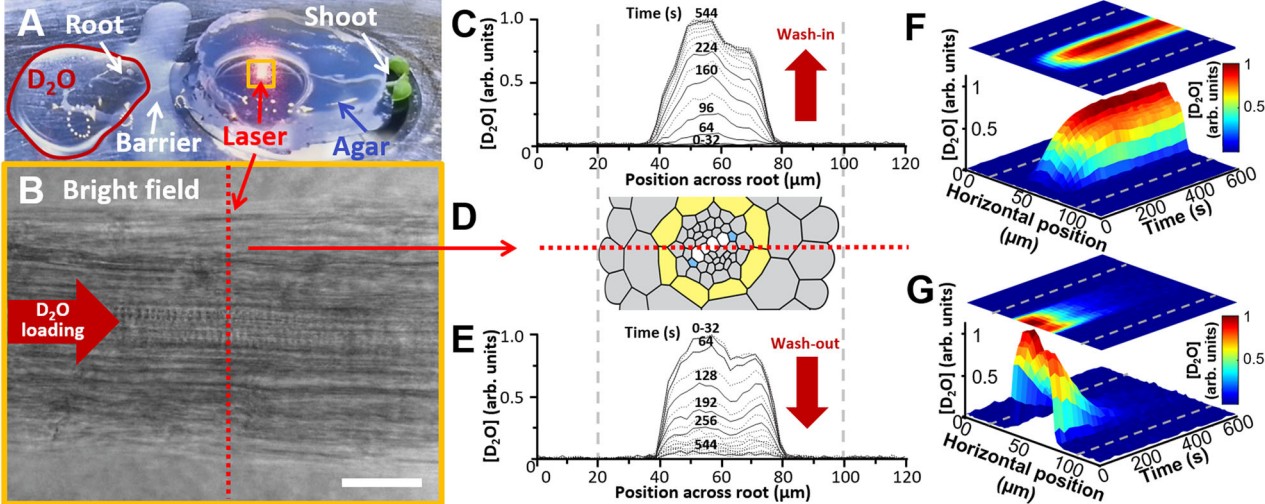

**Fig. 1 Overview of the water imaging system and its outputs in line-scan mode. A** Experimental setup for imaging hydrodynamics, showing *A. thaliana* under Raman measurement (laser). $D_2O$ loaded via the intact root tip is separated from the rest of the plant by a silicone grease barrier. The $H_2O$-containing agar cap placed over the root prevents movement and desiccation. The location of panel B is indicated by the orange box. **B** Bright-field micrograph of intact root (scale bar 25 μm). The dotted red line indicates the location of line scans in (**C–G**). Note the visible xylems in this central plane of focus. **C** Time-course of $D_2O$ wash-in during successive line scans across the root center. Timings (in seconds) define the start of each line scan. **D** Root anatomy, showing the location of the confocal RMS line scan (**C, E**) across the full diameter, through the slightly tilted stele, with highlighted endodermis (yellow), protoxylem vessels (blue), and immature xylem (white). Spatial resolution is diffraction-limited at a 2 μm step size, temporal resolution is 300 ms/point (detector limit). **E** Time-course of $D_2O$ wash-out during successive line scans. Timings (in seconds) define the start of each line scan. **F** 3D plot showing spatiotemporal dynamics of $D_2O$ pulse arrival at the line under observation. **G** 3D plot showing spatiotemporal dynamics of $D_2O$ washing out of the preparation when $H_2O$ is re-introduced via the intact root. Spatial resolution is diffraction-limited at a 2 μm step size. Temporal resolution is 300 ms/point (detector limit). Dotted gray lines in **C–G** indicate the outer boundary of the root under observation. Source data underlying Fig. 1C, E–G are provided as a Source Data file.

parameters (Supplementary Table 2). To better constrain the estimation of hydraulic conductivity parameters during this first step, we complemented the $Lp_r$ data in WT, *CDEF*, and *sgn3-3 myb36-2* with data from the literature[19] of the well-characterized *esb1.1 CDEF* mutant line which shows ectopic lignification in endodermal radial walls, absent suberization, and downregulated aquaporins. In a second step, diffusion parameters and xylem water flow rates were estimated in another inverse modeling scheme, this time aimed at reproducing measured xylem $D_2O$ wash-out traces in each individual replicate plant (overview of distributions in Fig. 2F and individual fits in Supplementary Figs. 5–7).

MECHA simulations suggest that the main accelerator of the $D_2O$ wash-out in Fig. 2F is the advective water uptake in the distal region, close to the tip. This seems to be because $H_2O$ flushes out the remaining $D_2O$ reserves from outer root tissues in the distal root region previously immersed in $D_2O$. In WT, due to the presence of both endodermal barriers close to the RMS observation point (but not in the distal region), most of the uptake is distal and hence evacuates $D_2O$ reserves relatively quickly during the wash-out phase. To reach similar or steeper wash-out slopes as observed in WT (Fig. 2F), the endodermal barrier mutants thus required xylem water flow rates significantly higher than in WT (Fig. 2G, Anova-1 $p$-value $< 10^{-3}$, see individual values in Supplementary Table 2). This result corroborates consistently higher stomatal aperture in *CDEF* relative to WT observed independently[19], naturally generating higher xylem water flow rates.

In conclusion, this non-destructive hydrodynamic imaging approach produces meaningful quantitative results and parameters, supported by the concurrence of both stomatal and RMS observations. Furthermore, our imaging approach and model predictions suggest water transported via the root xylem does not re-enter outer root tissues or the surrounding soil when *en-route* to shoot tissues in *A. thaliana* plants with an intact endodermal diffusion barrier, thus distinguishing 'two water worlds'.

## Methods

**Raman microspectroscopy.** Raman spectra were recorded with a bespoke Raman microspectrometer optimized for biological samples. The instrument consists of a Raman laser (Tsunami 3960, Spectra-Physics) fitted with a 60 ps GTI and tuned at 725 nm wavelength to minimize sample autofluorescence while maximizing the Raman signal. The laser beam is passed through a clean-up filter (FF01-720/13-25, Laser2000) then directed inside an inverted microscope frame (Nikon Ti-E Eclipse), via the backport, using a dichroic mirror (FF735-DiO1-25 × 36, Semrock). A silver mirror (21010, ChromaTech) inside the microscope filter cassette is used to direct the laser into a water-immersion objective (63×/NA 1.0; Zeiss). Laser power at the sample was approximately 150 mW with a 2.2 μm laser spot size. Cells irradiated by lasers in the near-infrared region (700–1100 nm) remain viable under longer exposures at much higher laser power densities than in our experiment[20]. Raman photons collected by the microscope objective are passed through a long pass high-Q filter (HQ735LP, ChromaTech) upon exiting the backport of the microscope and focused using a 25 mm focal lens (AC127-025-B-ML, Thorlabs) into an optical fiber (WF50/125A 0.12NA, Ceramoptec). The optical fiber is connected to a spectrometer (77200, Oriel), equipped with an 830 lines/mm grating and a cooled deep-depletion back-illuminated CCD detector (iDus401, Andor). A high-precision automated stepper-motor stage (Prior, Cambridge, UK) fitted to the microscope is used to provide automated XY positioning. The wave-number axis of the spectrometer was calibrated using a NeAr atomic line source (IntelliCal, Princeton Instruments) and the spectral resolution was ~3 cm$^{-1}$ in the 2000–3000 cm$^{-1}$ region. Purpose-designed stainless steel microscope slides were fabricated that incorporated a Raman-silent $MgF_2$ window (0.11 mm thick, 15 mm diameter) to enable the acquisition of Raman spectra.

For measurements, a freshly cleaned microscope slide (stainless steel with an $MgF_2$ window insert) was prepared in advance by placing a water-proof barrier layer using a small syringe filled with silicone grease (Techne Ltd., part number 6101351). The barrier was dispensed onto the stainless steel part of the slide very close to the $MgF_2$ window. A few drops of water were added on both sides of the barrier to prevent the dehydration of plants. Plant samples were placed on the slide with the leaves on the $MgF_2$ side of the barrier and the root tip on the stainless steel side of the barrier (Fig. 1A). Then, another layer of the water-proof barrier was added on top of the root, overlapping the first barrier layer to completely seal the root. Finally, an agar cap was placed over the root on the $MgF_2$ section to prevent sample movement and root dehydration during the measurements. For the $D_2O$ wash-in experiments, the water ($H_2O$) on the stainless steel side of the grease barrier was replaced with $D_2O$, while for the $D_2O$ wash-out experiments the $D_2O$ was replaced with $H_2O$. Room temperature was 20 °C, relative humidity 43%, and light intensity 100 μmol m$^{-2}$ s$^{-1}$, which were all kept constant across experiments.

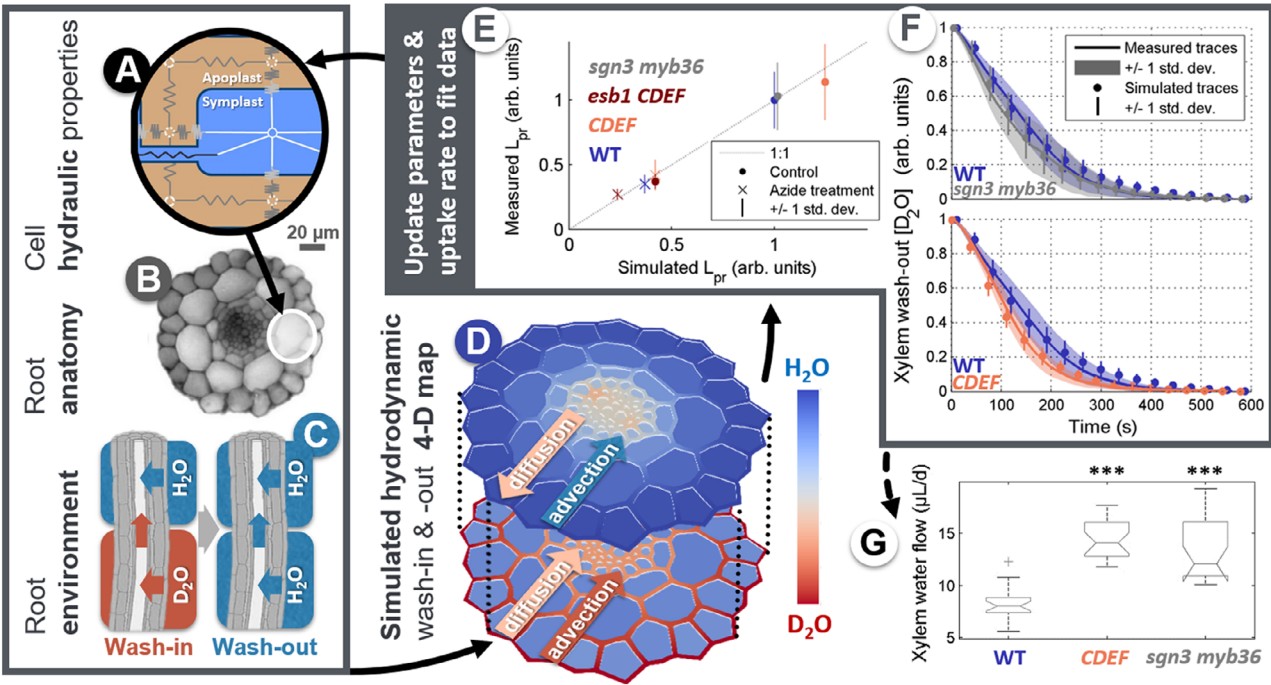

**Fig. 2 Scheme of the inverse modeling loop used to extract quantitative xylem water flow rate data from RMS D2O wash-out traces. A** Scheme of the subcellular compartments and pathways of the hydraulic model MECHA. **B** *A. thaliana* root anatomy used in the model. **C** Scheme of $D_2O$ wash-in and wash-out conditions. **D** Snapshot of a simulated $D_2O$ map for a mutant with no Casparian strip during the wash-in phase. **E** Fit of measured and simulated root hydraulic conductivities used to constrain the cell hydraulic parameters (WT: $N = 15$; CDEF: $N = 12$; esb1 CDEF: $N = 10$; sgn3 myb36: $N = 10$ independent plants; vertical bars show the extent of one standard deviation on both sides of the mean values). **F** Fit of measured and simulated $D_2O$ wash-out traces (WT: $N = 24$; CDEF: $N = 18$; sgn3 myb36: $N = 9$ independent plants; vertical bars and shaded areas show the extent of one standard deviation on both sides of the mean values). **G** Retrieved xylem water flow rates (WT: $N = 24$; CDEF: $N = 18$; sgn3 myb36: $N = 9$ independent plants; boxplots show 25th, 50th, and 75th percentiles, whiskers the most extreme points excluding outliers "+"). Triple stars indicate Anova-1 p-values below 0.001. Source data underlying Fig. 2E–G are provided as a Source Data file.

*Line scanning (Fig. 1).* Raman spectra were measured at different positions inside the plant by raster-scanning the root across the diffraction-limited laser focus in 2 μm steps (equivalent to a line of 60 points). This process was repeated 60 times (across the same line) in order to capture the temporal evolution of $D_2O$ during the wash-in and wash-out processes. The acquisition time at each position was 300 ms, yielding a total measurement time of approximately 32 min (includes raster scanning overhead). The same sample could be reproducibly scanned through multiple imaging cycles without visible photodamage. Data pre-processing consisted of three steps: cosmic ray removal, background subtraction, and $D_2O$ peak area integration. A very small fraction of individual spectra contained cosmic rays (typically < 1%). These are identified and the individual points (contaminated by cosmic rays) inside the Raman spectrum are discarded and replaced with the average value of the neighboring points. The average of the Raman spectra measured at points outside the plant represents the background spectrum (contribution from the agar, $MgF_2$, coverslip, and microscope objective). The Raman spectrum representing each point inside the plant was obtained by algebraic subtraction of the background spectrum. Finally, the area of the $D_2O$ peak in the 2200–2800 $cm^{-1}$ region was estimated using a Gaussian peak fit with a linear baseline. The obtained value is proportional to the $D_2O$ content at each point inside the plant.

*Time-course (Fig. 2F).* Raman spectra were measured at the same position inside the xylem (roughly 6–9 mm away from the root-hypocotyl junction) for approximately 16 min (1000 s), acquiring one measurement per second using kinetic mode acquisition inside the Andor SOLIS software. To confirm sample viability, each sample was pre-loaded with $D_2O$ via the root tip and placed on the microscope. During the $D_2O$ loading phase (approximately 15 min) a suitable measurement site inside the xylem was identified and the microscope focus was adjusted to maximize the Raman signal. After the loading phase was complete (no further increase in the Raman signal); $D_2O$ was replaced with $H_2O$ at the root tip and the $D_2O$ wash-out measurement was performed. Once complete, $H_2O$ was replaced with $D_2O$ and the wash-in measurement was performed. Data preprocessing consisted of four steps: cosmic rays removal, laser power fluctuation correction, background subtraction, and $D_2O$ peak area integration. Laser power fluctuation correction is performed for each individual Raman spectrum by normalizing every point inside the spectrum to the average value of all Raman points in the 2800–3000 $cm^{-1}$ region. The background spectrum (contributions from the agar, $MgF_2$ coverslip, microscope objective, and the plant itself) is calculated by averaging the last 20 spectra (981–1000 s) in the case of $D_2O$ wash-out, or by averaging the first 10 spectra (0–9 s) for the case of $D_2O$ wash-in. The

Raman spectrum representative of each point during the time course was obtained by algebraic subtraction of the background spectrum. Finally, the transient $D_2O$ profile was obtained by using Gaussian peak fitting similar to Fig. 1.

**Plant material, growth conditions, and $Lp_r$ measurements.** The accession used for all experiments in this study is *A. thaliana* Columbia-0 (Col-0). Plants were required for three types of experiments: Raman imaging of water transport, root anatomical imaging, and root hydraulic measurements. As the last two methods are invasive, these were conducted on different plants. Only plant material used for root anatomical imaging was fixed. For Raman imaging of water transport, seeds were surface-sterilized with 50% (v/v) sodium hypochlorite and 0.01% (v/v) Triton X-100, washed three times with sterile water, and sown on plates with 0.5× Murashige and Skoog (MS) medium (Sigma) solidified with 1% (w/v) Bactoagar (Difco). After two days at 4 °C in the dark, plates were placed vertically in a growth room at 21 °C with continuous light at 100–150 μmol $m^{-2}\,s^{-1}$. To obtain root sections used as templates for MECHA, seeds were treated as above but grown at 22 °C.

Root hydraulic measurements were performed with pressure chambers on 21-day-old plants cultivated for 10 days in vitro and an additional 11 days in hydroponic solution in a 16 h photoperiod growth chamber at 21/20 °C. Root hydraulic conductivity ($Lp_r$) was determined in freshly detopped roots using a set of pressure chambers filled with a hydroponic culture medium. Excised roots were sealed using dental paste (Coltène/Whaledent s.a.r.l., France) and were subjected to 350 kPa for 10 min to achieve flow stabilization, followed by successive measurements at pressures 320, 160, and 240 kPa. Root hydrostatic conductance ($K_r$) was calculated by the slope of the flow-to-pressure relationship. The hydraulic conductivity was calculated by dividing $K_r$ by the root dry weight. For sodium azide (NaN₃) experiments, $Lp_r$ were determined from continuous measurement at 320 kPa. In order to compare measured and simulated $Lp_r$ expressed as hydraulic conductances per dry weight and per root surface area, respectively; values were normalized by the mean $Lp_r$ of the WT plants under control conditions, hence the arbitrary units in Fig. 2E and Supplementary Fig. 4.

**Root anatomical imaging.** In order to preserve accurate cell geometry, roots were fixed before acquiring and extracting the anatomy from microscope images. Root samples were fixed for 3 h at 20 °C in a solution containing 4% glutaraldehyde, 4% formaldehyde, and 50 mM sodium phosphate buffer at pH 7.2. Serial ethanol

dehydration was then performed (30, 50, 70, 90, and 95% [twice]) at room temperature for 1 h at each step. Samples were embedded in Technovit 7100 resin (Kulzer) according to the manufacturer's instructions. Sections were cut, dried onto glass slides, and stained for 20 min in an aqueous 0.1% calcfluor solution. Sections were observed under a fluorescence microscope equipped with a UV filter set.

**Solution of solute transport equations from the subcellular scale**. The version of the micro-hydrological model MECHA[17] used in this study solves three-dimensional transient solute transport in finite-difference hydraulic networks with the implicit Crank–Nicolson method[21]. The hydraulic networks comprise symplastic and apoplastic "nodes", where solute concentration is defined. Nodes are connected by hydraulic conductances (Fig. 2A) conveying water and solutes through paths including cell walls, plasma membranes, and plasmodesmata. Solutes may move along these paths with water mass flow (advection) or due to random molecular movements from a region of high to a region of low concentration (diffusion). In case the solute is deuterated water ($D_2O$), no substantial metabolism, exclusion, or active transport applies so that advection and diffusion are the only relevant processes to consider when simulating its transport. For more elaborate simulations of the transport of major ions and associated processes in *A. thaliana* roots, an axisymmetric radial-longitudinal model has been developed by Foster and Miklavcic[22,23]. In MECHA, primary cell walls, plasma membranes, plasmodesmata, and xylem vessels are attributed to specific diffusion coefficients and hydraulic conductivities. The diffusivity of $D_2O$ within the protoplast is considered as non-limiting given its high porosity relative to membranes and plasmodesmata so that $D_2O$ concentration is assumed to freely equilibrate and thus to be uniform within each protoplast. In consequence, each protoplast only needs a single node at its center, while the apoplast is discretized in apoplastic blocks, according to cell wall geometry (parts of walls between two adjacent cells form one of these blocks). Apoplastic nodes are located at the center of each apoplastic block and at their junctions (see Fig. 2A, dotted and dashed white circles, respectively) for a total of 517 nodes in the apoplast and symplast within each two-dimensional plane in this study. In addition to connections between nodes within each transverse plane, apoplastic nodes from consecutive planes are axially connected through cell wall hydraulic conductances, while symplastic nodes are axially connected through two conductances in parallel standing for (i) plasmodesmata, and (ii) a cell wall and two plasma membrane layers. Unlike other apoplastic nodes connected by hydraulic conductances representative of primary (permeable) or secondary (hydrophobic) walls, the axial conductances of xylem vessels are calculated with the Poiseuille–Hagen law, based on their diameters determined from root sections[24].

In this study, we focus on regions of the root upstream of the observation point which substantially affects xylem water composition. We, therefore, exclude the elongation zone, which does not have functional xylem vessels and radially absorbs part of the water it needs for cell elongation. For this reason, and because roots stayed fully hydrated during experiments, we assumed that cell volumes represented in the three-dimensional modeling framework did not change over time. Consequently, the root hydraulic anatomy has constant volumes for pieces of compartments, with specific diffusivities at their interfaces. An *ad-hoc* version of the solute advection-diffusion equation accounting for these specificities is solved. It relies on (i) the calculation of solute flow rates between compartments, (ii) the solute mass balance in each compartment, and (iii) solute boundary conditions at nodes in direct contact with the root environment. In the following elaboration, time (e.g., day), length (e.g., meter), and quantity (e.g., mole) units are referred to with the T, L, and N symbols, respectively. For enhanced clarity, scalars are represented by symbols in italics, vectors by symbols in bold italics, and matrices by symbols in bold.

As common plant root tissues do not fractionate $H_2O$ and $D_2O$[25], solute advective flow rate ($Q_{s,a}$, $N\,T^{-1}$) across each path simply equals the product of water flow rate (($Q_w$, $L^3\,T^{-1}$) by $D_2O$ concentration ($C$, $N\,L^{-3}$) at the origin (node $i$ in Eq. (1), node $j$ in Eq. (2))

$$Q_{s,a,i\rightarrow j} = Q_{w,i\rightarrow j}C_i \tag{1}$$

or

$$Q_{s,a,i\leftarrow j} = Q_{w,i\leftarrow j}C_i \tag{2}$$

where the subcellular water flow rate between nodes $i$ and $j$ is previously solved as in Couvreur et al.[17].

Unlike solute advective flow, solute diffusive flow rate ($Q_{s,d}$, $N\,T^{-1}$) does not depend on the direction of water flow. It is driven by the difference of solute concentration between neighboring compartments:

$$Q_{s,d,i,j} = D_p \frac{A_{i,j}}{l_{i,j}}(C_i - C_j) \tag{3}$$

where $D_p$ ($L^2\,T^{-1}$) is the $D_2O$ effective diffusion coefficient in path type $p$, $A_{i,j}$ ($L^2$) and $l_{i,j}$ ($L$) are the cross-section and length of the path connecting node $i$ and its neighboring node $j$, respectively.

In order to track $D_2O$ concentration change in time, solute advective and diffusive flow rates are balanced in the following equation for each node

$$V_i \frac{\partial C_i}{\partial t} = -\sum_{\text{outflow}} Q_{s,a,i\rightarrow j} + \sum_{\text{inflow}} Q_{s,a,i\rightarrow j} - \sum_j Q_{s,d,i,j} \tag{4}$$

where $V_i$ ($L^3$) is the fixed volume of the cell solution allocated to node $i$, outgoing (respectively incoming) solute advective flow rates negatively (respectively positively) contribute to the solute mass in compartment $i$, and solute diffusive flow rates are summed over all connections with neighboring compartments ($j$).

A consequence of the $V_i$ factor on the left-hand side of Eq. (4) is the buffering of solute concentration in compartments with high volumes (e.g., cortical protoplast), while concentration may fluctuate faster in compartments with low volumes (e.g., cell walls). Based on Gaff and Carr[26], water is assumed to occupy 69% of the primary cell wall volume and 70% of the protoplast volume.

Finally, two types of Neumann boundary conditions are set in Eq. (4) for nodes at the root surface and at the proximal end of xylem vessels. First, a solute advective inflow or outflow rate term equal to the product of water flow rate by the associated solute concentration at the origin (e.g., 0 for root surface walls bathing in $H_2O$; equal to water inflow rate for root surface walls bathing in $D_2O$, then subtracted from the boundary condition vector $F_{BC}$ at the row corresponding to the node, see Eq. (6)). Second, a solute diffusive flow rate term, which is a function of the concentration difference between the chosen boundary condition and the root surface wall. The latter term may accelerate the propagation of the solute front and tends to zero as the compartment's concentration equilibrates with the boundary concentration.

Equations (1–4) are solved with the implicit Crank–Nicolson method[21], which uses the following approximation to ensure the convergence and accuracy of the solution

$$\frac{C_{t+dt} - C_t}{dt} \simeq \frac{1}{2}\frac{\partial C_t}{\partial t} + \frac{1}{2}\frac{\partial C_{t+dt}}{\partial t} \tag{5}$$

where $C_t$ and $C_{t+dt}$ ($N_{tot} \times 1$, $N\,L^{-3}$) are vectors of solute concentrations at all root nodes at times $t$ and $t + dt$, respectively ($dt$ being a relatively small time step). $N_{tot}$ is the total number of nodes (specific nodes indices are referred to as $i$ or $j$ in Eqs. (1–4)). In order to solve Eq. (5), one must express $t$ as a function of $t + dt$, then regroup it to the left-hand side. Here, this is possible because the water flow field is at steady-state at least from time $t$ to $t + dt$ (note that $D_2O$ concentration does not need to be at steady-state, so the relation between $C$ and its temporal derivative is known, see Eq. (6)).

Equations (1–4) can be compacted into the following matrix operation

$$\text{diag}(V) \cdot \frac{\partial C}{\partial t} = M \cdot C - F_{BC} \tag{6}$$

where $V$ ($N_{tot} \times 1$, $L^3$) is the vector of cell solution volumes associated to each root node (associated to the specific node $i$ in Eq. (4)), $V$ (Ntot $\times N_{tot}$, $L^3$) is a matrix containing the vector $V$ on its central diagonal and zeros elsewhere, $M$ ($N_{tot} \times N_{tot}$, $L^3\,T^{-1}$) contains the factors multiplying solute concentrations on the right hand side of Eqs. (1–3), while Neumann solute boundary conditions are concatenated in $F_{BC}$ ($N_{tot} \times 1$, $N\,T^{-1}$).

The matrix $M$ is built as follows: for each node $i$, the outgoing water flow rates $Q_{w,i\rightarrow j}$ are subtracted from $M[i,j]$ (first coordinate for row index, second coordinate for column index) and incoming water flow rates $Q_{w,i\leftarrow j}$ from each neighboring node $j$ added to $M[i, j]$; for each node $i$, the "diffusion connectivity" $D_p \frac{A_{i,j}}{l_{i,j}}$ with each neighboring node $j$ is subtracted from $M[i, i]$ and added to $M[i, j]$.

Combining Eqs. (5) and (6), $C_t$ and $C_{t+dt}$ being two distinct versions of the vector $C$, yields the following matrix operation

$$(\text{diag}(V) - \frac{dt}{2} \cdot M) \cdot C_{t+dt}(\text{diag}(V) - \frac{dt}{2} \cdot M) \cdot C_t - dt \cdot F_{BC} \tag{7}$$

In this system, after setting initial concentrations in the vector $C_t$, the remaining unknowns are grouped in $C_{t+dt}$. The system is solved with the Python Scipy function "spsolve" for sparse matrices.

**Model parametrization**. The root hydraulic layout reproduces the anatomy of an *A. thaliana* primary root in the differentiation zone, with a mature protoxylem (Fig. 2B, Supplementary Fig. 2). The skeleton of the hydraulic anatomy can be digitized with the software CellSet[27], as in the current study (version 1.5.1 used here), or simulated with the root anatomy simulator GRANAR[28]. As axial water fluxes through phloem sieve tubes are relatively small compared to xylem water fluxes, here we assume phloem flow does not affect the xylem $D_2O$ content at the RMS measurement point. Hence, for simplicity, phloem elements and their companion cells are treated as stele parenchyma cells in the simulations of $D_2O$ advection-diffusion. The cross-section is given a third spatial dimension, assuming $10^{-4}$ m long cells[29] stacked axially (here we tested the model with up to 280 stacks since the longest distance between the start of the differentiation zone and the laser focal point was about 2.8 cm). As the focus of this study extends to regions of the root that transport water toward the RMS measurement point, MECHA explicitly simulates $D_2O$ advection-diffusion from the transverse section located at the maturation point of protoxylem vessels onto the RMS measurement point, about 2 cm closer to the shoot.

Two types of hydrophobic barriers are modeled in the endodermal cell walls of the selected lines. First, the CS—made of lignin— blocking apoplastic flow in radial cell walls of endodermal cells, already at the point of maturation of protoxylem vessels in WT. This barrier is absent up to 3 mm from the protoxylem maturation point in the mutant *esb1.1* (and by extension *esb1.1 CDEF*, whose hydraulic data is

used to constrain the model parameters) before an ectopic CS is formed[19]. The CS is completely absent in *sgn3-3 myb36-2*, even after the formation of the suberin lamellae, so that apoplastic flow in radial walls of endodermal cells may continue at all stages of maturation[16]. Second, the suberin lamellae, made of a waxy substance called suberin, is located between plasma membranes and primary walls on all sides of endodermal cells. Therefore, it blocks water flow between the apoplast and the symplast of the endodermis (short dark gray resistances in Fig. 2A). Suberin lamellae fully cover the endodermis about 1 cm after the CS in *A. thaliana* WT[19]. The suberin-degrading mutant *CDEF* (and by extension *esb1.1 CDEF*) does not display an endodermal suberin lamellae[19]. To isolate the impact of hydrophobic barrier locations on $D_2O$ dynamics, and as no substantial difference in root anatomies were observed between WT and mutants, the same network geometry is used for all simulations. Thus, simulated variations between different lines only stem from cell hydraulic properties adjusted at hydrophobic barrier locations as summarized in Supplementary Table 1.

Water flow and diffusion in the $10^{-7}$ m thick *A. thaliana* cell walls[30] are limited by their hydraulic conductivity and diffusion coefficient. Assuming that they are fully hydrophobic, CS and suberin lamellae are attributed null hydraulic conductivities and diffusion coefficients, so they locally block any $D_2O$ flux. The primary cell wall hydraulic conductivity and diffusion coefficient that allow the best fits of $Lp_r$ (Fig. 2E) and wash-out traces data (Fig. 2F) are $2.5 \times 10^{-12}$ m$^2$ MPa$^{-1}$ s$^{-1}$ and $9.4 \times 10^{-10}$ m$^2$ s$^{-1}$, respectively.

Advective water flow through non-suberized cell plasma membranes is known to occur through protein channels called aquaporins[31] as well as a small part through the phospholipid bilayer of cell membranes. The phospholipid bilayer hydraulic conductivity is set to $2.6 \times 10^{-8}$ m MPa$^{-1}$ s$^{-1}$, assuming similar physical properties as in Couvreur et al.[17], while the contribution of aquaporins to the plasma membrane hydraulic conductivity that allows the best fit of $Lp_r$ data (Fig. 2E) is $5.4 \times 10^{-7}$ m MPa$^{-1}$ s$^{-1}$, except in *esb1.1 CDEF*, whose aquaporin contribution is fivefold lower due to downregulated aquaporins[19]. $D_2O$ diffusion across the phospholipid bilayer of cell plasma membranes is also modeled, with a diffusion coefficient of $3.4 \times 10^{-17}$ m$^2$ s$^{-1}$ after multiplication by the partition coefficient of water in the lipid bilayer[32,33], whose thickness is only $3.0 \times 10^{-9}$ m[34].

Advection and diffusion within the symplast occur through plasmodesmata, whose open cross-section is set to $7.5 \times 10^{-5}$ m$^2$ over a length equivalent to twice the thickness of a primary cell wall ($2 \times 10^{-7}$ m)[35]. Based on Zhu et al.[36], plasmodesmata frequencies are set to tissue-specific values reported in Supplementary Table 3.

The hydraulic conductance per individual plasmodesma and associated diffusion coefficient that allow the best fits of the $Lp_r$ (Fig. 2E) and wash-out traces data (Fig. 2F) are $1.1 \times 10^{-19}$ m$^3$ MPa$^{-1}$ s$^{-1}$ and $8.9 \times 10^{-10}$ m$^2$ s$^{-1}$, respectively.

The last media to have its own hydraulic conductivity and diffusion coefficient is the xylem vessel. The specific hydraulic conductance of xylem poles calculated with Poiseuille–Hagen law is $2.1 \times 10^{-14}$ m$^4$ MPa$^{-1}$ s$^{-1}$ and the diffusion coefficient that allows the best fit of the wash-out traces data (Fig. 2F) is $1.2 \times 10^{-9}$ m$^2$ s$^{-1}$, which is relatively low and barely affects the simulated traces in xylem vessels. As a matter of comparison, the self-diffusion coefficient of $D_2O$ in free water at 25 °C approaches $2.3 \times 10^{-8}$ m$^2$ s$^{-1}$ [37,38]. Diffusion coefficients with similar values to those we found have been reported for small ionic compounds such as Na$^+$ and K$^+$ in *A. thaliana* cell walls[23], which we interpret as a positive sign in view of the large uncertainties associated with the estimation of diffusion coefficients in plant tissues[39]. Apart from the hydraulic conductivity of membranes, which matches the range found in maize[40], the subcellular hydraulic conductivities of cell walls and plasmodesmata are smaller than those found in maize with the same modeling framework[17,41]. Reduced diffusion coefficients and hydraulic conductivities in *A. thaliana* porous media are likely due to lower porosity, constrictivity, and higher tortuosity[42].

Though the model parameters are all physical, no direct method allows measuring them easily, simultaneously, or non-invasively. Inverse modeling is an indirect method allowing the retrieval of model parameter values by searching for the values that best reproduce measured variables that can be simulated by the model. In this study, the measured variables are the root hydraulic conductivity ($Lp_r$) and the xylem $D_2O$ washout trace. Different parameter values are tested as part of a "search loop" (Fig. 2A–F) until simulations and measurements converge.

With the search algorithm Multistart from the MATLAB (MathWorks, Inc.) global optimization toolbox, we find parameter values of primary cell wall hydraulic conductivity ($k_{PD}$), individual plasmodesma hydraulic conductance ($K_{PD}$), and aquaporin contribution to plasma membrane hydraulic conductivity ($k_{AQP}$) that best fit the $Lp_r$ data of WT, *esb1.1 CDEF*, *CDEF*, and *sgn3-3 myb36-2*, under control and azide treatments, except for *sgn3-3 myb36-2* for which only control $Lp_r$ data is available (Fig. 2E, optimal parameter values in Supplementary Table 2). As the measured $Lp_r$ data has units of water flow rate per pressure differential per root mass, while $Lp_r$ simulated with the 3-D version of MECHA has units of water flow rate per pressure differential per root surface, all $Lp_r$ values are normalized by the WT $Lp_r$ under control treatment (leaving six other data points to constrain three unknowns). Note that $Lp_r$ simulations include the region of the root starting at the point of maturation of protoxylem vessels and consecutive 1.5 cm (150 stacks) shootward.

The same principle is used to find parameter values for the diffusion coefficients of primary cell walls ($D_w$), plasmodesmata ($D_{PD}$), xylem vessels ($D_x$), and for the xylem axial water flow rate associated to each trace ($Q_{xyl}$). Overall, simulated $D_2O$

dynamics are quite sensitive to the type of apoplastic barrier and to $Q_{xyl}$, which has two consequences: (i) measured $D_2O$ traces can be expected to be sensitive to xylem water flow rate, which is a requisite for our enterprise of quantifying such fluxes in living tissues based on $D_2O$ traces, and (ii) in plants with different apoplastic barriers, similar $D_2O$ traces may have been driven by substantially different values of xylem water flow rates (as confirmed in Fig. 2F, G) and vice versa. Absolute values of $D_2O$ content would be hard to capture experimentally due to variations in the instrument transfer function between samples and fluctuations in laser power over the weeks to months during which the experiments took place. Therefore, we focus on reproducing the shape of the normalized wash-out traces (linearly transformed to start at a value of one, and end at a value of zero, 600 s after the start of the wash-out phase, Fig. 2F). Interestingly, unlike absolute fractions of $D_2O$ in simulated wash-out traces, the normalized curves do not vary substantially when scaling root length with xylem water flow rate (see Supplementary Fig. 8) while the sensitivity to the type of apoplastic barrier remains (see Supplementary Fig. 9). Thus, in order to save computing time and resources, the inverse modeling loop is conducted on a shorter virtual root of 0.15 cm (15 stacks), with 0.05 cm exposed to $D_2O$ during the wash-in phase (replaced by $H_2O$ during the wash-out phase), 0.05 cm through the barrier separating distal from proximal pools of $D_2O/H_2O$, and 0.05 cm in the proximal $H_2O$ pool. The distribution of measured and simulated wash-out traces (average and ±1 standard deviation area) are shown in Fig. 2F, as well as individually in Supplementary Figs. 5–7. Optimal parameter values that allowed reaching these fits are displayed in Supplementary Table 2. Note that as there is experimental data for 51 wash-out traces, and 54 wash-out parameters (the diffusion coefficients being conserved across simulated traces), the average number of parameters per experimental trace is 1.06.

**Reporting summary**. Further information on research design is available in the Nature Research Reporting Summary linked to this article.

## Data availability

Additional data supporting the findings of this work are available within the paper and its Supplementary Information files. A reporting summary for this Article is available as a Supplementary Information file. The root hydraulic conductivity data used in this study have been deposited in the FigShare database [https://doi.org/10.6084/m9.figshare.14815911.v2].

The Raman microspectroscopy data generated in this study have been deposited in the FigShare database [https://doi.org/10.6084/m9.figshare.14815971]. Source data are provided with this paper.

## Code availability

The latest code of MECHA working in 3D with solute advection-diffusion and associated Matlab codes for inverse modeling schemes are openly available online under GPL.2 open-source licence at FigShare [https://doi.org/10.6084/m9.figshare.14892408.v2] or GitHub [https://github.com/MECHARoot/MECHA/blob/master/MECHA_4Dsolute. zip]. The custom code used to acquire and analyze the Raman microspectroscopy data is openly available online at the FigShare database [https://doi.org/10.6084/m9. figshare.14815671]. The commercial softwares ANDOR SOLIS (4.2230007.0), ANDOR SDK (2.94.30007.0), and Prior scientific software Version 1.78.0.0 were also used in this study to acquire the Raman microspectroscopy data.

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

# ARTICLE

8.  Borisjuk, L., Rolletschek, H. & Neuberger, T. Surveying the plant's world by magnetic resonance imaging. *Plant J.* **70**, 129–146 (2012).
9.  Windt, C. W., Vergeldt, F. J., De Jager, P. A. & Van As, H. MRI of long-distance water transport: a comparison of the phloem and xylem flow characteristics and dynamics in poplar, castor bean, tomato and tobacco. *Plant Cell Environ.* **29**, 1715–1729 (2006).
10. Hayat, F., Zarebanadkouki, M., Ahmed, M. A., Buecherl, T. & Carminati, A. Quantification of hydraulic redistribution in maize roots using neutron radiography. *Vadose Zone J.* **19**, e20084 (2020).
11. Puppels, G. J. et al. Studying single living cells and chromosomes by confocal Raman microspectroscopy. *Nature* **347**, 301–303 (1990).
12. Doblas, V. G., Geldner, N. & Barberon, M. The endodermis, a tightly controlled barrier for nutrients. *Curr. Opin. Plant Biol.* **39**, 136–143 (2017).
13. Von Wangenheim, D., Goh, T., Dietrich, D. & Bennett, M. J. Plant biology: building barriers… in roots. *Curr. Biol.* **27**, R172–R174 (2017).
14. Deng, J., Xiong, T., Wang, H., Zheng, A. & Wang, Y. Effects of cellulose, hemicellulose, and lignin on the structure and morphology of porous carbons. *ACS Sustain. Chem. Eng.* **4**, 3750–3756 (2016).
15. Naseer, S. et al. Casparian strip diffusion barrier in Arabidopsis is made of a lignin polymer without suberin. *Proc. Natl Acad. Sci. USA* **109**, 10101–10106 (2012).
16. Reyt, G. et al. Two chemically distinct root lignin barriers control solute and water balance. *Nat. Commun.* **12**, 2320 (2021).
17. Couvreur, V. et al. Going with the flow: multiscale insights into the composite nature of water transport in roots. *Plant Physiol.* **178**, 1689–1703 (2018).
18. Tournaire-Roux, C. et al. Cytosolic pH regulates root water transport during anoxic stress through gating of aquaporins. *Nature* **425**, 393–397 (2003).
19. Wang, P. et al. Surveillance of cell wall diffusion barrier integrity modulates water and solute transport in plants. *Sci. Rep.* **9**, 4227 (2019).
20. Neuman, K. C., Chadd, E. H., Liou, G. F., Bergman, K. & Block, S. M. Characterization of photodamage to Escherichia coli in optical traps. *Biophys. J.* **77**, 2856–2863 (1999).
21. Perrochet, P. & Bérod, D. Stability of the standard Crank-Nicolson-Galerkin Scheme applied to the diffusion-convection equation: some new insights. *Water Resour. Res.* **29**, 3291–3297 (1993).
22. Foster, K. J. & Miklavcic, S. J. A comprehensive biophysical model of ion and water transport in plant roots. II. Clarifying the roles of SOS1 in the salt-stress response in arabidopsis. *Front. Plant Sci.* **10**, 1121 (2019).
23. Foster, K. J. & Miklavcic, S. J. A comprehensive biophysical model of ion and water transport in plant roots. I. Clarifying the roles of endodermal barriers in the salt stress response. *Front. Plant Sci.* **8**, 1326 (2017).
24. Heymans, A., Couvreur, V. & Lobet, G. Combining cross-section images and modeling tools to create high-resolution root system hydraulic atlases in *Zea mays*. *Plant Direct*. e334 https://doi.org/10.1002/pld3.334 (2021).
25. Dawson, T. E. & Ehleringer, J. R. Streamside trees that do not use stream water. *Nature* **350**, 335–337 (1991).
26. Gaff, D. F. & Carr, D. T. The quantity of water in the cell wall and its significance. *Aust. J. Biol. Sci.* **14**, 299–311 (1961).
27. Pound, M. P., French, A. P., Wells, D. M., Bennett, M. J. & Pridmore, T. P. CellSeT: novel software to extract and analyze structured networks of plant cells from confocal images. *Plant Cell* **24**, 1353–1361 (2012).
28. Heymans, A., Couvreur, V., LaRue, T., Paez-Garcia, A. & Lobet, G. GRANAR, a computational tool to better understand the functional importance of monocotyledon root anatomy. *Plant Physiol.* **182**, 707–720 (2020).
29. De Cnodder, T., Vissenberg, K., Van Der Straeten, D. & Verbelen, J.-P. Regulation of cell length in the *Arabidopsis thaliana* root by the ethylene precursor 1-aminocyclopropane- 1-carboxylic acid: a matter of apoplastic reactions. *N. Phytol.* **168**, 541–550 (2005).
30. Andème-Onzighi, C., Sivaguru, M., Judy-March, J., Baskin, T. I. & Driouich, A. The reb1-1 mutation of Arabidopsis alters the morphology of trichoblasts, the expression of arabinogalactan-proteins and the organization of cortical microtubules. *Planta* **215**, 949–958 (2002).
31. Chaumont, F. & Tyerman, S. D. Aquaporins: highly regulated channels controlling plant water relations. *Plant Physiol.* **164**, 1600–1618 (2014).
32. Khakimov, A. M., Rudakova, M. A., Doroginitskii, M. M. & Filippov, A. V. Temperature dependence of water self-diffusion through lipid bilayers assessed by NMR. *Biophysics* **53**, 147–152 (2008).
33. Mathai, J. C., Tristram-Nagle, S., Nagle, J. F. & Zeidel, M. L. Structural determinants of water permeability through the lipid membrane. *J. Gen. Physiol.* **131**, 69–76 (2007).
34. Lewis, B. A. & Engelman, D. M. Lipid bilayer thickness varies linearly with acyl chain length in fluid phosphatidylcholine vesicles. *J. Mol. Biol.* **166**, 211–217 (1983).
35. Ehlers, K. & Kollmann, R. Primary and secondary plasmodesmata: structure, origin, and functioning. *Protoplasma* **216**, 1–30 (2001).
36. Zhu, T., Lucas, W. J. & Rost, T. L. Directional cell-to-cell communication in the Arabidopsis root apical meristem I. An ultrastructural and functional analysis. *Protoplasma* **203**, 35–47 (1998).
37. Baur, M. E., Garland, C. W. & Stockmayer, W. H. Diffusion coefficients of $H_2O$-$D_2O$ mixtures. *J. Am. Chem. Soc.* **81**, 3147–3148 (1959).
38. Liu, H. & Macedo, E. A. Accurate correlations for the self-diffusion coefficients of $CO_2$, $CH_4$, $C_2H_4$, $H_2O$, and $D_2O$ over wide ranges of temperature and pressure. *J. Supercrit. Fluids* **8**, 310–317 (1995).
39. Kramer, E. M., Frazer, N. L. & Baskin, T. I. Measurement of diffusion within the cell wall in living roots of *Arabidopsis thaliana*. *J. Exp. Bot.* **58**, 3005–3015 (2007).
40. Ehlert, C., Maurel, C., Tardieu, F. & Simonneau, T. Aquaporin-mediated reduction in maize root hydraulic conductivity impacts cell turgor and leaf elongation even without changing transpiration. *Plant Physiol.* **150**, 1093–1104 (2009).
41. Ding, L. et al. Modification of the expression of the aquaporin ZmPIP2;5 affects water relations and plant growth. *Plant Physiol.* **182**, 2154–2165 (2020).
42. Grathwohl, P. *Diffusion in Natural Porous Media: Contaminant Transport, Sorption/Desorption and Dissolution Kinetics*. ISBN 978-0-7923-8102-0 https://link.springer.com/book/10.1007/978-1-4615-5683-1 (1998).

## Acknowledgements

This work was supported by awards from the Biotechnology and Biological Sciences Research Council (grant Nos. BB/K010212/1 to K.F.W., BB/T001437/1 to M.J.B./D.M.W., BB/V003534/1 to D.E.S./M.J.B., BB/L027739/1, and BB/N023927/1 to D.E.S.) The Leverhulme Trust (RPG-2016-409, to M.J.B./D.M.W./K.F.W.), and from the ERA-NET Coordinating Action in Plant Sciences program (ERACAPS13.089_RootBarriers to DES). V.C. was funded by as Research Fellow by the Belgian Fonds de la Recherche Scientifique (F.R.S.-FNRS, grant number: 1208619F), M.C.-P. has received funding from the EU's Seventh Framework Program under grant agreement N° FP7-609398 (AgreenSkills + contract), I.C. was funded by Junta de Extremadura Spain (GR18168). K.F.W. was funded by a Royal Academy of Engineering/EPSRC Postdoctoral Fellowship (EP/G058121/1).

## Author contributions

M.J.B. and K.F.W. designed and supervised the project. F.C.P. and K.F.W. conceived RMS study and design. F.C.P. designed and built RMS instrumentation. F.C.P. and D.D. carried out RMS experiments. V.C. designed the model and inverse modeling program. V.C., F.C.P., M.J.B. and K.F.W. co-wrote the paper. N.L., G.R., Y.B., M.C.-P., I.C., C.M., D.E.S., X.D. and D.M.W. contributed to the discussion and revision of the paper.

## Competing interests

The authors declare no competing interests.
