## [Peer Review File · Nature Communications]

REVIEWER COMMENTS

Reviewer #1 (Remarks to the Author):

This paper reported a study using Raman microscopy to image the fluxes of water using D₂O, and analyzed using hydrodynamics modelling.

The non-invasive capability of Raman microscopy is well-explored in many applications. From the technology perspective, it is not new, but the application is original.

I'm not the expert in the hydrodynamics modelling, so I'm not able to comment on that.

The main concern/question regarding Raman microscopy is the laser power they used is very high (150 mW), any tissue damage observed? Especially concerned when repeated measurement on the same position was conducted.

line 242: "root sections". Why root sample needs fixation? Will the fixation influence the hydrodynamic imaging?

In addition, There are many grammatical mistakes.

Reviewer #2 (Remarks to the Author):

The authors of the manuscript "Non-invasive hydrodynamic imaging in plant roots at cellular resolution" report the use of Raman microspectroscopy, in combination with hydrodynamic modelling, as promising new tools to monitor and predict water fluxes through plant tissues at cellular scale with high temporal resolution.

While Raman microspectroscopy has previously been used to probe chemical composition of different plant tissues, it is exciting to see that this technique can also be used to gain a better understanding of water transport in plant tissues non-invasively.

After a brief introduction, the manuscript is divided into two parts corresponding to Figures 1 and 2.

Figure 1 gives a clear overview of the experimental setup and the results that have been obtained. It is very impressive to see how efficiently D₂O is contained by the endodermal layer. There are only a few minor points that should be addressed:

- 1.) In Figure 1B a box could be added to mark the position where the D₂O was applied. From the text it appears that D₂O was applied at the root tip. A visual element could help readers to understand the experiment more easily from the figure.
- 2.) L76,77: How was the time of 80 seconds determined? In Figure 1D it appears that D₂O was already detected after 30 seconds. It would be useful to show the background signal over time and show when the difference became significant.

3.) How many replicates were done for this experiment (if any)? I assume the flux of D₂O would depend on the conditions on the microscope stage (i.e. temperature, relative humidity and light exposure of the shoot). It would be useful to add additional information regarding this in the method section.

4.) L87: Remove the parentheses around the reference.

Additionally, I have a couple of questions:

1.) I'm wondering why the word "convection" is used throughout the manuscript (e.g. L84). In my opinion you're describing mass flow in most cases and this is the common term that is used in the literature to describe this process. Convection as the combination of mass flow and diffusion would only apply if there was also a concentration gradient. Please explain why you chose the term convection. Otherwise, I would suggest to replace "convection" with the term "mass flow".

2.) You conclude in lines 86 to 90 that water is not likely to diffuse from the stele to the cortex. I'm wondering if your observation could be influenced by the direction of flow in the system. If the Casparian strip (CS) was porous and, hence, permeable to water, water molecules may flow from the cortex into the stele due to the pressure gradient. This may counteract diffusion in the other direction. I'm wondering if the system has been observed long enough to allow time for diffusion and if stopping the transpiration stream (for example by submerging or applying silicone grease to the shoot), which would stop mass flow, would lead to a gradual diffusion over time.

Figure 2 shows how D₂O wash-out experiments and modelling, using a new implementation of MECHA, were used to determine xylem flow rates in wild-type plants, Casparian strip, and suberin mutants. The combination of these techniques appears very useful, but is not easy to understand right away. I think that the explanation of the model could be improved through modification of Figure 2 or an additional supplementary figure. Relating to this and other points of this section, I have the following comments:

1.) From the figure it is not very clear how anatomy, hydraulic conductivities, and wash-out traces come together to inform the model. I think adding a more detailed explanation of this figure, for example to the supplementary materials, would help readers to better understand how the model works. For example the description of the model from line 250 could be illustrated as well.

2.) Figure 2E: I understand that the hydraulic conductivities were measured to test if the simulated values are accurate. It would be good to point this out in the text.

3.) Figure 2E: Why was it necessary to include the esb1 CDEF mutant? Please explain.

4.) Figure 2E: Why was azide treatment not performed for the sgn3 myb36 mutant?

5.) Figure 2F: How many measurements were performed; i.e. number of replicates?

6.) L111: Please add the original reference for the pCASP1::CDEF construct as well.

7.) L134-136, Fig.S3: Comparing actual measurements, similar to Fig. 1D and E, for these mutants to the simulations would demonstrate the validity of the model. This is particular important due to the claim in lines 159-162.

8.) L155: I think higher root hydraulic conductivity for the CDEF mutant (shown in Fig. 2E) could also lead to higher flow rates and possibly cause positive feedback in stomatal conductance.

Reviewer #3 (Remarks to the Author):

Summary

The manuscript is a report on a combined experimental and theoretical study of water transport in roots of *Arabidopsis thaliana*. The method utilizes Raman (micro) spectroscopy for the non-invasive capture of deuterated water movement in living root tissue. The method relies on the inverse mapping of a convective-diffusive transport model to deduce dynamic parameters/properties of root tissue elements. The study includes a sequence of experiments wherein roots are exposed to deuterated water alternatively normal water to study rates of transport into and through wildtype and mutant roots of *Arabidopsis*.

Comments

The material presented is a novel application of theory and experiment and represents an important methodological advance on the in situ and non-invasive study of water movement through plants. It would seem to offer a significant possibility to quantify important tissue-specific transport properties. While the method still seems a long way off being available to the “common” plant biologist, it is still an important step in that direction.

I think the paper could be published in Nature Communications after the authors have taken the following particular comments into account.

1. The sentences in Lines 45-47 seem out of place with the rest of the document. Issues with water resources are not addressed in this report and there is no mention of the connection with the study of water movement in plants. I would suggest either removing or bringing these sentences into the fold.
2. The sentence in Lines 49-51 seems incomplete.
3. I would suggest that the sentence beginning with “In this work ..” in line 64 should begin a new paragraph.
4. Line 67 “were” should read “is”.
5. Line 70-71, the use of parentheses is inconsistent.
6. Line 87, reference 11 is placed in parentheses but no where else is this done.
7. Line 106, should maxima be maximum?
8. Line 146, “aiming” should be “aimed”.
9. The sentence beginning with “RMS” in Line 107 could possibly begin a new paragraph.

10. Despite the large number of authors the paper suffers from an all-too frequent language hiccup which interrupts the flow. Although in the above comments I have pointed out a few such in the main document, the issues arise mostly in the methods section. I would suggest another careful look by the native English speakers.
11. The Main Text section appears to end rather abruptly. Was this intended? The last paragraph of this section begins with a presentation of results, whereas it (sort of) ends with some semi-summary statement. Perhaps the paragraph could be split into two with the second being an encapsulating conclusion paragraph.
12. In the Methods section describing the theory, why are the words convection, diffusion and connectivity in quotation marks (Lines 256, 257 and 329)?
13. Line 263, I imagine what is implied is that the D₂O concentration is assumed uniform in the protoplasts. If so, perhaps you should make that explicit.
14. In Equation 4 shouldn't the left hand side of the equality read $\sum_{i=1}^{N_{tot}} X_i$ to account for possible cell volume changes? Or, are volumes assumed fixed?
15. In Lines 282,286,293,297,299,319,323,326,328, and possibly in other places "I" appears where "t" is meant - probably a MSWord autocorrect error.
16. In Equation 5 the right hand side requires knowledge of future values of C_t and τ . Could the authors explain this conundrum?
17. I was a little confused by the similar notation for node value X_i and vector C_t . Could these be distinguished in a more distinctive way?
18. In Eq 6, I was also confused by the inconsistency of left and right hand terms of the equation. On the right the two terms represent $N_{tot} \times 1$ vectors, while the right hand side appears to be a $N_{tot} \times N_{tot}$ matrix. This is probably just a poorly expressed way of writing what is meant, but could the authors fix it?
19. In Line 345 et seq, how many stacks are assumed in the model? It is not clear from my reading of the manuscript.
20. Generally, I kept forgetting what RMS stood for, I kept thinking of root mean square, which is what RMS typically stands for. Perhaps the authors could use something different such as R_{oCS} ?
21. In reading through the description of the theory I was struck by the essential similarity with the model presented by Foster and Miklavcic (Frontiers in Plant Science 2018, 2019, 2021). Although the model here utilizes a root cross section to set up the root (as stacks of cells with this cross section and of 10^{-4} m length) rather than circular arc cylinder segments, as imagined by Foster and Miklavcic, the entire model of discrete equations is identical (naturally), making the calculations effectively the same. So too is the idea of optimizing parameter set against experimental data to deduce physically realistic parameters. The comparison is further enhanced by the fact that Foster and Miklavcic also model an *Arabidopsis thaliana* root. I would have thought it appropriate to mention this earlier work and comment on likenesses or differences. How do the optimized parameter values compare? Are the differences in specific model features important? Are similar traits found and behaviour found?

22. Figures 1 and 2 are not so easy to read. Indeed in Figure 1 (D-H) it is difficult to read the figure axes (titles and values). The same applies to Figure 2 (F-G). Also, Figure 1 (A) is rather difficult to make out, despite the labelling. While I understand the origin of Figure 1 (C), I am not sure I understand what I am looking at. Is it a longitudinal section of the root, or is it just a enlarged view of the root exterior surface. I would appreciate a revision of these figures to make them more legible and easier to comprehend.

Recommendation

I recommend acceptance subject to minor revision.

In this document, comments by reviewers appear in black and **replies by the authors in blue**. Line numbers in blue between brackets refer to the resubmitted version manuscript 'without track-change', but the manuscript 'with track-change' comparing the initial submission and current resubmission of the manuscript is also shared for convenience in case the referees would like to consult it.

Reviewer #1 (Remarks to the Author):

This paper reported a study using Raman microscopy to image the fluxes of water using D₂O, and analyzed using hydrodynamics modelling. The non-invasive capability of Raman microscopy is well-explored in many applications. From the technology perspective, it is not new, but the application is original. I'm not the expert in the hydrodynamics modelling, so I'm not able to comment on that.

We thank the referee for the positive comment on the originality of this RMS application and for pointing out the required clarifications.

- 1) The main concern/question regarding Raman microscopy is the laser power they used is very high (150 mW)

- a. any tissue damage observed?

No, we observed no photodamage in roots. We were able to follow multiple wash-in / wash-out cycles on the same plant (see also Reviewer 2, point 3; in comments related to Fig. 1).

- b. Especially concerned when repeated measurement on the same position was conducted.

While living cells are photosensitive, the near-infrared is well known to be comparatively gentle on cells. We are using 725nm laser light, and plant cells seem robust to this in our hands. Similar laser power regimes have been used previously in the study of embryoid bodies with no influence on the ability of human embryonic stem cells (hESC) to differentiate into cardiac phenotype within embryoid bodies¹. These delicate structures are considerably more photosensitive than plant cells appear to be, and yet were also unperturbed.

We have clarified these important points in the manuscript as follows:

"Cells irradiated by lasers in the near-infrared region (700 nm-1100 nm) remain viable under longer exposures at much higher laser power densities than in our experiment" [L 189-91]

"The same sample could be reproducibly scanned through multiple imaging cycles without visible photodamage." [L 219-20]

Below we have included a technical relation to published literature for editorial surety, though we feel its inclusion would bloat the text:

It is well known that biomolecules are sensitive to light, and exposure of cells to lasers can cause localized heating or induce toxic photochemical reactions. Laser wavelengths in the visible range (400-600nm) can induce biological cell damage even at low laser powers and short exposure time (Puppels, G. J., J. H. Olminkhof, J. Greve. 1991. *Laser irradiation and Raman spectroscopy of single living cells and chromosomes: sample degradation occurs with 514.5 nm but not with 660 nm laser light. Exp. Cell Res.* 195:361-367.). In contrast, when cells are irradiated by lasers in the near-infrared region (700nm-1100nm) they can remain viable even after longer exposures at much higher laser power densities. Dr Pascut has used 150-170mW of power at 785nm extensively in his study of mammalian stem cell differentiation processes using Raman microscopy. For instance embryoid bodies were irradiated with such laser powers for 1 to 3h each day for 9 days

with no influence on the ability of hESCs to differentiate into the (beating) cardiac phenotype (Pascut, F. C., H. T. Goh, V. George, N. Welch, C. Denning, and I. Notingher. 2013. *Non-invasive label-free monitoring the cardiac differentiation of human embryonic stem cells in-vitro by Raman spectroscopy. Biochimica et Biophysica Acta. 1830:3517-3524*). No plant tissue damage was observed at any time during our measurements in this study.

2) line 242: "root sections". Why root sample needs fixation?

We needed the transverse root cell geometry in order to simulate hydrodynamics in *A. thaliana* roots accurately with MECHA. To obtain high quality resin sections, pre-fixation is necessary to preserve the cell geometry. This was clarified in the Methods section of the revised document as follows:

"In order to preserve accurate cell geometry, roots were fixed before acquiring and extracting the anatomy from microscope images." [L 264-5]

Will the fixation influence the hydrodynamic imaging?

Root fixation and non-invasive hydrodynamics imaging are two separate parts of the experiment conducted on separate plants. Fixation was solely used to preserve the anatomy for the fluorescence imaging to guide the model (L 330-338), while hydrodynamics were necessarily measured in unfixed living plants – indeed fixation would irrevocably destroy the processes of interest. This is explicit in the methods section of the revised document as follows:

"Plants were required for three types of experiments: Raman imaging of water transport, root anatomical imaging, and root hydraulic measurements. As the last two methods are invasive, these were conducted on different plants. Only plant material used for root anatomical imaging was fixed." [L 250-3]

In addition, there are many grammatical mistakes.

We apologize for this, and have ensured their elimination from the revised document.

Reviewer #2 (Remarks to the Author):

The authors of the manuscript "Non-invasive hydrodynamic imaging in plant roots at cellular resolution" report the use of Raman microspectroscopy, in combination with hydrodynamic modelling, as promising new tools to monitor and predict water fluxes through plant tissues at cellular scale with high temporal resolution. While Raman microspectroscopy has previously been used to probe chemical composition of different plant tissues, it is exciting to see that this technique can also be used to gain a better understanding of water transport in plant tissues non-invasively.

We would like to thank the referee for the apposite summary and the enthusiastic comment about our novel application of RMS to directly measure water transport in plant tissues non-invasively.

After a brief introduction, the manuscript is divided into two parts corresponding to Figures 1 and 2. Figure 1 gives a clear overview of the experimental setup and the results that have been obtained. It is very impressive to see how efficiently D₂O is contained by the endodermal layer. There are only a few minor points that should be addressed:

1.) In Figure 1B a box could be added to mark the position where the D₂O was applied. From the text it appears that D₂O was applied at the root tip. A visual element could help readers to understand the experiment more easily from the figure.

This has been implemented, as a demarcation line around the D₂O drop rather than a simple box (which we found distracting). Figure 1B was eliminated as redundant and the caption reflects this. Signposting via colored lines and arrows now assists the reader to visualize where D₂O was applied, and where the scanning took place in 3D.

The new version of appears on p4, and is reproduced in our reply to the final comment of referee #3 below. We thank the reviewer for suggesting this and find this version clearer.

2.) L76,77: How was the time of 80 seconds determined?

It was calculated as follows: first two scans showed no change in D₂O (32+32=64s) the first time we see an obvious drop in D₂O was in the 3rd line scan, at position 50 μm- 13 s into the scan. Therefore 64+13= 77 s, which was rounded up to 80 seconds.

Figure 1D is appears that D₂O was already detected after 30 seconds It would be useful to show the background signal over time and show when the difference became significant.

This appears to have been an inadvertent mis-reading of an overly-crowded draft panel. Panels C&E have consequently been redrafted in this revision to clearly show the departure from baseline pertaining to wash-in/wash-out as the 64s line scan – now indicated through clearly labelled solid lines. Intervening traces are dotted to minimize visual clutter. The first line scan in which the signal clearly departs from each baseline begins at 64 s in both cases, however as calculated in (2) above the first relevant timepoint captured inside the stele (where the signal appears confined) is closer to 80s after exposure of the root tip to D₂O. We believe that this is now clear in the revised version of the manuscript, which clarifies in Fig. 1 caption that the times indicated above each curve is the start time of the line scan:

“(C) Time-course of D₂O wash-in during successive line scans across the root center. Timings (in seconds) define the start of each line scan.”

“(E) Time-course of D₂O wash-out during successive line scans. Timings (in seconds) define the start of each line scan.”

3.) How many replicates were done for this experiment (if any)?

Line scans in this representative experiment was repeated for four imaging cycles (two wash-in and two wash-out) at the same location within the same plant. Each cycle included about 20 line scans, so there were approximately 80 line scans in total. All confirmed the absence of D₂O in outer root tissues, while the D₂O peak was clearly visible within the stele. Subsequent data in Fig 2 was acquired at a single location within the xylem to parameterize the MECHA model with high temporal resolution. In the latter experiment, there were 24, 18 and 9 replicates in WT, *CDEF* and *sgn3 myb36*, respectively. These pieces of information are now explicit in the revised version of the manuscript:

“Spatially, the D₂O pulse was detected solely in inner root tissues (i.e. the endodermis and the stele, which contains water-transporting xylem vessels) and not in outer root tissues, at any point of a 20-minute experiment composed of wash-in and wash-out phases (Fig. 1C & E).” [L 80-82]

“For example, the *CDEF* line needed the shortest time to complete D₂O wash-out (linear slope of the wash-out curve: $-5.2 \times 10^{-3} \pm 0.7 \times 10^{-3} \text{ s}^{-1}$, N=18, significantly steeper than in WT: $-3.9 \times 10^{-3} \pm 0.9 \times 10^{-3} \text{ s}^{-1}$, N=24, Anova-1 p-value < 10^{-4}), whereas *sgn3-3 myb36-2* had almost no effect on the wash-out slope ($-4.4 \times 10^{-3} \pm 1.1 \times 10^{-3} \text{ s}^{-1}$, N=9) compared to WT (Fig. 2F, where traces are normalized to start at a unit value).” [L 122-6]

I assume the flux of D₂O would depend on the conditions on the microscope stage (i.e. temperature, relative humidity and light exposure of the shoot). It would be useful to add additional information regarding this in the method section.

The reviewer is correct that factors influencing transpiration rate (temperature, relative humidity and light exposure) would indeed likely alter measured D₂O fluxes. For this reason these factors were kept constant throughout experiments (temperature: 20° C ; RH 43% ; Light intensity: 100 μmol m⁻² s⁻¹), to avoid the introduction of variability other than due to the *sgn3.3 myb36.2* mutation and *CDEF* construct. These environmental variables are explicitly quantified in the revised version of the manuscript. We do agree that elegant physiology could be carried out in future by manipulation of these key parameters, but this is beyond the scope of this paper.

“Room temperature was 20° C, relative humidity 43%, and light intensity 100 μmol m⁻² s⁻¹, which were all kept constant across experiments.” [L 212-3]

4.) L87: Remove the parentheses around the reference.
Corrected in the revised version of the manuscript.

Additionally, I have a couple of questions:

1.) I’m wondering why the word “convection” is used throughout the manuscript (e.g. L84). In my opinion you’re describing mass flow in most cases and this is the common term that is used in the literature to describe this process. Convection as the combination of mass flow and diffusion would only apply if there was also a concentration gradient. Please explain why you chose the term convection. Otherwise, I would suggest to replace “convection” with the term “mass flow”.

We thank the referee for noting this potential point of semantic confusion in the term “convection”. In the emerging field of root micro-hydrological modelling, which simulates both the movement of solutes with solvent drag and its diffusion down the solute concentration gradient at the sub-cellular scale, the term “convection” has been commonly used for the mass flow (i.e. advective) component of solute transport ²⁻⁴. However, following the referee’s advice, we checked that in some other fields it appears that the term “convection” is used to describe the resultant of both mass flow (advection) and diffusion. Because in hydrological transport modelling the terms “convection-diffusion equations” and “advection-diffusion equations” are most commonly referred to, and because the term “advection” appears to be more consentient ^{5,6}, we have adopted the term “advection” in place of “convection” throughout the manuscript. The manuscript was revised accordingly, and the variable $Q_{s,c}$ was renamed $Q_{s,a}$.

We also noted a need for the clarification that there does exist a concentration gradient between H₂O and D₂O, which drives their interdiffusion due to random molecular agitation. This is why we include a diffusion term in equations of D₂O transport. This is now clarified in the *Methods* section.

2.) You conclude in lines 86 to 90 that water is not likely to diffuse from the stele to the cortex. I'm wondering if your observation could be influenced by the direction of flow in the system. If the Casparian strip (CS) was porous and, hence, permeable to water, water molecules may flow from the cortex into the stele due to the pressure gradient. This may counteract diffusion in the other direction. I'm wondering if the system has been observed long enough to allow time for diffusion and if stopping the transpiration stream (for example by submerging or applying silicone grease to the shoot), which would stop mass flow, would lead to a gradual diffusion over time.

Based on our estimations, the D₂O diffusivity in hydrated primary cell walls (D_w) is about $9.4 \cdot 10^{-10} \text{ m}^2\text{s}^{-1}$. Assuming a unit D₂O gradient from one side to the other of the endodermis cell layer, which is about 5 μm wide, we estimate that without a Casparian strip the diffusive flux of D₂O in radial cell walls of the endodermis would be about $200 \mu\text{m s}^{-1}$. This is more than 100 times the typical advective water flux estimated in cell walls, thus radially D₂O diffusion seems to dominate advection in cell walls. In this hypothetical scenario, with no apoplastic barrier, it would thus take 0.025 s for D₂O to reach the cortex. Assuming a diffusivity 100 times lower across the Casparian strip, due to its lower porosity, it would still only take 2.5 seconds for D₂O to reach the cortex. Cumulatively, a wash-in and wash-out cycle in our experiments lasted more than 1200 seconds (20 minutes), so we are convinced that outer root tissues were observed for long enough to exclude D₂O permeation through the endodermal apoplastic barrier.

Figure 2 shows how D₂O wash-out experiments and modelling, using a new implementation of MECHA, were used to determine xylem flow rates in wild-type plants, Casparian strip, and suberin mutants. The combination of these techniques appears very useful, but is not easy to understand right away. I think that the explanation of the model could be improved through modification of Figure 2 or an additional supplementary figure. Relating to this and other points of this section, I have the following comments: 1.) From the figure it is not very clear how anatomy, hydraulic conductivities, and wash-out traces come together to inform the model. I think adding a more detailed explanation of this figure, for example to the supplementary materials, would help readers to better understand how the model works.

We thank the reviewer for the insightful comment, which we have opted to embody conceptually as the graphical abstract for the manuscript. This suggestion was extremely useful to crystallize our thinking on this important communication route – we believe this significantly improves the immediacy of reader appreciation of the paper as a whole and thank the reviewer again for their excellent encapsulation of its salience. The following Graphical abstract will be advertised with the manuscript upon publication:

This addition was complemented by two new figures in supplementary materials (Supplementary Fig. 7-8), briefly explained in the Methods section as follows:

“Overall, simulated D_2O dynamics are quite sensitive to the type of apoplastic barrier and to Q_{xyl} , which has two consequences: (i) measured D_2O traces can be expected to be sensitive to xylem water flow rate, which is a requisite for our enterprise of quantifying such fluxes in living tissues based on D_2O traces, and (ii) in plants with different apoplastic barriers, similar D_2O traces may have been driven by substantially different values of xylem water flow rates (as confirmed in Fig. 2F,G) and vice versa. Absolute values of D_2O content would be hard to capture experimentally due to variations in the instrument transfer function between samples and fluctuations in laser power over the weeks to months during which the experiments took place. Therefore, in the following, we focus on reproducing the shape of the normalized wash-out traces (linearly transformed to start at a value of one, and end at a value of zero, 600 seconds after the start of the wash-out phase, Fig. 2F). Interestingly, unlike absolute fractions of D_2O in simulated wash-out traces, the normalized curves do not vary substantially when scaling root length with xylem water flow rate (see Supplementary Fig. 7) while the sensitivity to the type of apoplastic barrier remains (see Supplementary Fig. 8). Thus, in order to save computing time and resources, the inverse modelling loop is conducted on a shorter virtual root of 0.15 cm (15 stacks), with 0.05 cm exposed to D_2O during the wash-in phase (replaced by H_2O during the wash-out phase), 0.05 cm through the barrier separating distal from proximal pools of D_2O / H_2O , and 0.05 cm in the proximal H_2O pool.” [L 458-74]

Supplementary Figure 7. Sensitivity analysis of D_2O wash-out traces simulated with MECHA to (i) the length of the distal part of the root previously immersed in D_2O (see symbols in the legend), and (ii) to

the distance between the laser focal point and the grease barrier separating the proximal and distal parts of the root (see color code in the legend). A: Absolute fractions of D₂O in xylem water are sensitive to (i) and (ii). B: Normalized fractions of D₂O in xylem water are neither sensitive to factors (i) nor (ii).

Supplementary Figure 8. Sensitivity analysis of D₂O wash-out traces simulated with MECHA to (i) the type of apoplastic barrier (see symbols in the legend), and (ii) to the distance between the laser focal point and the grease barrier separating the proximal and distal parts of the root (see color code in the legend). A: Absolute fractions of D₂O in xylem water are sensitive to factors (i) and (ii). B: Normalized fractions of D₂O in xylem water remain sensitive to factor (i) only.

For example the description of the model from line 250 could be illustrated as well.

We agree with the referee. In the revised version of the manuscript, we refer the reader to Fig. 2A for a detailed illustration of the different types of apoplastic nodes in the root hydraulic anatomical network:

“Apoplastic nodes are located at the center of each apoplastic block and at their junctions (see Fig. 2A, dotted and dashed white circles, respectively) for a total of 517 nodes in the apoplast and symplast within each two-dimensional plane in this study.” [L 290-3]

2.) Figure 2E: I understand that the hydraulic conductivities were measured to test if the simulated values are accurate. It would be good to point this out in the text.

Actually, as the numerical values of the hydraulic conductances of cell walls, membranes and plasmodesmata are highly uncertain *a priori*, we needed a way to narrow down their three estimated parameter values. For that we used an approach called “inverse modelling”, which takes advantage of the fact that L_{p_r} data is available and can be simulated with the model, to search for the set of these three uncertain parameter values that best reproduce measured L_{p_r} data. This was clarified in the revised version of the manuscript as follows:

“MECHA can simulate D₂O spatio-temporal dynamics during wash-in and wash-out cycles (Fig. 2C-D) in WT, CDEF, and *sgn3-3 myb36-2* hydraulic anatomies under physiological conditions (e.g. snapshots at the laser focal point before wash-out starts in Supplementary Fig. 3, showing D₂O leakage in *sgn3-3 myb36-2* due to the absence of the Casparian strip). Conducting such simulations first required the estimation of hydraulic conductivity and diffusivity parameters values using an inverse modelling

scheme based on an iterative search algorithm (loop indicated by curved solid arrows between panels in Fig. 2) that fine-tunes these values until convergence between measured and simulated variables. The convergence between root hydraulic conductivities (termed L_{p_r}) simulated and measured with a pressure chamber⁷ under control and azide treatments (inactivating aquaporins⁸)(Fig. 2E) drove the search algorithm to optimal values for the three sub-cellular hydraulic conductivity parameters (Supplementary Table 2).” [L 144-154]

3.) Figure 2E: Why was it necessary to include the *esb1 CDEF* mutant? Please explain.

This is a point that indeed needed clarification in the text. With L_{p_r} data in WT, *CDEF* and *sgn3.3 myb36.2* alone we had five data points (L_{p_r} data in *sgn3.3 myb36.2* under azide treatment is currently unavailable). We considered that five observations were too small a number of points to estimate three parameter values, due to possible issues of non-unicity (i.e. multiple sets of parameter values could yield similar L_{p_r} predictions in WT, *CDEF* and *sgn3.3 myb36.2* lines, but different wash-out traces; and we would not know which hydraulic parameters values were the most representative ones). Therefore, we added two more L_{p_r} points from the *esb1 CDEF* mutant, which is well characterized in the literature. Note that aquaporin downregulation in *esb1 CDEF* was previously estimated to approximate -80% in the literature, so it did not add an extra unknown in the inverse modelling process. This allowed our estimation of hydraulic conductivity parameters values to be better constrained. This was clarified in the revised version of the manuscript as follows:

“To better constrain the estimation of hydraulic conductivity parameters during this first step, we complemented the L_{p_r} data in WT, *CDEF* and *sgn3-3 myb36-2* with data from the literature⁹ of the well-characterized *esb1.1 CDEF* mutant line which shows ectopic lignification in endodermal radial walls, absent suberization, and downregulated aquaporins.” [L 154-8]

4.) Figure 2E: Why was azide treatment not performed for the *sgn3 myb36* mutant?

The referee is correct. The L_{p_r} data under control and azide treatments was collected in a project in which the new *sgn3 myb36* mutant became available only after other lines had been already characterized. Due to its late availability, only the L_{p_r} data under control conditions was characterized. Two studies^{9,10} showed a strong correlation between a reduction of aquaporin expression and the reduction of L_{p_r} , particularly for L_{p_r} -azide (as observed in the *esb1* mutant). In the *sgn3 myb36* mutant, aquaporin expression analysis and L_{p_r} measurements showed similar levels as in WT. This strongly suggests that L_{p_r} -azide is not affected in the *sgn3 myb36* mutant, making the evaluation of the impact of azide on aquaporin-facilitated radial water transport irrelevant in this mutant. L_{p_r} -azide was therefore not characterized in *sgn3 myb36*. Fortunately, we do not need this data point because we have enough data to constrain the hydraulic parameters values.

5.) Figure 2F: How many measurements were performed; i.e. number of replicates?

We are glad that the referee pointed this out. In the previously-submitted version numbers of replicates (WT: 24 ; *CDEF*: 18 ; *sgn3.3 myb36.2*: 9) could only be found indirectly as the numbers of Q_{xyl} values in Supplementary Table 2 and numbers of subplots Supplementary Fig. 4-6, but they were not mentioned explicitly in the text. We have corrected this in the revised version of the manuscript as follows:

“For example, the *CDEF* line needed the shortest time to complete D₂O wash-out (linear slope of the wash-out curve: $-5.2 \cdot 10^{-3} \pm 0.7 \cdot 10^{-3} \text{ s}^{-1}$, N=18, significantly steeper than in WT: $-3.9 \cdot 10^{-3} \pm 0.9 \cdot 10^{-3} \text{ s}^{-1}$, N=24, Anova-1 p-value $< 10^{-4}$), whereas *sgn3-3 myb36-2* had almost no effect on the wash-out slope ($-4.4 \cdot 10^{-3} \pm 1.1 \cdot 10^{-3} \text{ s}^{-1}$, N=9) compared to WT (Fig. 2F, where traces are normalized to start at a unit value).” [L 122-6]

And also revised in Fig. 2 caption:

“F Fit of measured and simulated D₂O wash-out traces. G Retrieved xylem water flow rates (WT: N=24; *CDEF*: N=18; *sgn3 myb36*: N=9; boxplots show 25th, 50th and 75th percentiles, whiskers the most extreme points excluding outliers ‘+’)”

6.) L111: Please add the original reference for the pCASP1::*CDEF* construct as well.

The reference was amended in the revised version of the manuscript when introducing the pCASP1::*CDEF* construct:

“a wildtype line expressing the pCASP1::*CDEF* construct which degrades endodermal suberin (termed *CDEF*)¹¹” [L 119-20]

7.) L134-136, Fig.S3: Comparing actual measurements, similar to Fig. 1D and E, for these mutants to the simulations would demonstrate the validity of the model. This is particularly important due to the claim in lines 159-162.

The referee points out a possible point of confusion, which requires clarification. Our measurements with line scans in WT show that D₂O does not diffuse from the stele to the cortex over timescales that would have allowed us to see D₂O there if the endodermal diffusion barriers were leaky. Our model, parametrized independently from line scan data using L_p and xylem D₂O wash-out traces, also predicts that D₂O largely diffuses throughout the stele but does not reach outer tissues, as confirmed in the direct measurements using RMS. However, when we suggest that water may diffuse out into cortical tissues in *sgn3 myb36* whose Casparian strip is leaky, this is solely based on model predictions (Supplementary Fig. 3). The limited number of *sgn3 myb36* mutant plants available did not allow us to conduct the line scan experiment to confirm the model prediction, so we can only conclude that this result is suggested by the model.

In the revised version of the manuscript, the sentence was therefore rephrased as follows:

“our novel imaging approach and model predictions suggest water transported via the root xylem does not re-enter outer root tissues or the surrounding soil when *en-route* to shoot tissues in *A. thaliana* plants with an intact endodermal diffusion barrier”. [L 175-7]

We remove the claim that the intact Casparian strip is a *requirement* for that (i.e. that water would diffuse out of the stele if the Casparian strip was leaky), since only model predictions suggest it.

8.) L155: I think higher root hydraulic conductivity for the *CDEF* mutant (shown in Fig. 2E) could also lead to higher flow rates and possibly cause positive feedback in stomatal conductance.

We are glad to have sparked thoughts of integrative understanding resulting from our novel methods, leading to a synthesis of biophysical processes in the roots and leaves via

the shoot, and thus a deeper understanding of hydrodynamics in living plants. Supp. Fig. 5A of Wang et al. (2019) shows that, indeed, stomatal aperture is higher in the *CDEF* genotype than in WT. Such a positive feedback is likely to happen, provided that water is not limiting and that no additional hydraulic or ABA signal overrides it. Hypotheses considered in ^{9,12} include the fact that due to the absence of suberin lamellae, ions such as K⁺ do not accumulate in leaves like they would do in WT, which hinders stomatal closure as guard cells rely on K⁺ to increase their turgidity. Further work and cross-confirmation using state of the art methods such as real-time stomatal conductance measurements is both warranted and necessary towards such a complete understanding, but is beyond the scope of this paper.

Reviewer #3 (Remarks to the Author):

Comments

The material presented is a novel application of theory and experiment and represents an important methodological advance on the *in situ* and non-invasive study of water movement through plants. It would seem to offer a significant possibility to quantify important tissue-specific transport properties. While the method still seems a long way off being available to the “common” plant biologist, it is still an important step in that direction. I think the paper could be published in Nature Communications after the authors have taken the following particular comments into account.

We thank the referee for the very relevant suggestions, which helped improve the quality of the manuscript, and for taking the time to point out grammatical mistakes, which have been corrected accordingly.

1. The sentences in Lines 45-47 seem out of place with the rest of the document. Issues with water resources are not addressed in this report and there is no mention of the connection with the study of water movement in plants. I would suggest either removing or bringing these sentences into the fold.

The referee makes a fair point. Addressing the roles and complexity of water fluxes in plant tissues would make more sense at the start of the introduction of this paper. It has therefore been rewritten as follows:

“Water plays an essential role as a solvent for nutrients, minerals and other biomolecules in plant tissues^{13,14}. To date, the inability to non-invasively image and quantify water transport directly within root tissues has been a key stumbling block for researchers seeking to understand hydrodynamics within living plant cells and tissues.” [L 45-8]

2. The sentence in Lines 49-51 seems incomplete.

This is correct, thank you for spotting it. We added the term “or” to mark the last element in the list of current limitations of water imaging techniques:

“Current techniques developed to monitor water uptake in roots either suffer from being indirect (tracking radiotracers¹⁵ or monitoring pressure¹⁶) or invasive (e.g. pressure chamber⁷, root and xylem pressure probes¹⁷, heat pulsing¹⁸).” [L 48-50]

3. I would suggest that the sentence beginning with “In this work..” in line 64 should begin a new paragraph.

The manuscript has been revised accordingly.

4. Line 67 “were” should read “is”.
The manuscript has been revised accordingly.

5. Line 70-71, the use of parentheses is inconsistent.
The referee is correct. The revised sentence now reads:
“Our study explores whether this novel approach can provide non-invasive measurements of water fluxes in living plant tissues at a cellular (2µm step size) spatial resolution and sub-second (0.3s) temporal resolution.” [L 67-70]

6. Line 87, reference 11 is placed in parentheses but no where else is this done.
Formatting here is a bit tricky. The *Nature* formatting uses numbers as exponents to refer to cited articles in the reference section. When a citation is adjacent to a number, its units, or the formula of a chemical compound, the citation could be confused for an exponent in the adjacent formula. In the manuscript, we thought that putting the citation number between parentheses under these conditions would clarify that it is not a numerical exponent. This is a practice that we observed in a couple of Nature papers, and thought it would be sensible to apply this practice in our manuscript. We used it in the following instances: “the lignin pore diameter is ~1 nanometer (¹⁹)”; “plant root tissues do not fractionate H₂O and D₂O (²⁰)”; “whose thickness is only 3.0 10⁻⁹ m (²¹)”; “whose open cross-section is set to 7.5 10⁻⁵ m² (²²)”. We do not know what would be the best practice, and would yield to the copy-editing team if there are alternative ways to best mitigate these instances.

7. Line 106, should maxima be maximum?
The referee is correct. There are two protoxylem vessels displaying local maxima of D₂O, but in this particular sentence it is correct to refer to “a single protoxylem vessel (identified as a local maximum of D₂O (...)).” [L 115]
This has been corrected accordingly in the revised manuscript.

8. Line 146, “aiming” should be “aimed”.
The manuscript has been revised accordingly.

9. The sentence beginning with “RMS” in Line 107 could possibly begin a new paragraph.
The manuscript has been revised accordingly.

10. Despite the large number of authors the paper suffers from an all-too frequent language hiccup which interrupts the flow. Although in the above comments I have pointed out a few such in the main document, the issues arise mostly in the methods section. I would suggest another careful look by the native English speakers.
We apologize for the residual grammatical mistakes. The revised version of the document has been proof-read by multiple native English speakers.

11. The Main Text section appears to end rather abruptly. Was this intended? The last paragraph of this section begins with a presentation of results, whereas it (sort of) ends with some semi- summary statement. Perhaps the paragraph could be split into two with the second being an encapsulating conclusion paragraph.
The referee makes a good suggestion. We did not intend to give the impression of an abrupt end of the Main Text section, and have adjusted the concluding statement in a separate paragraph, as suggested by the referee. It now reads:

“In conclusion, this novel, non-destructive hydrodynamic imaging approach produces meaningful quantitative results and parameters, supported by the concurrence of both stomatal and RMS observations. Furthermore, our novel imaging approach and model predictions suggest water transported via the root xylem does not re-enter outer root tissues or the surrounding soil when *en-route* to shoot tissues in *A. thaliana* plants with an intact endodermal diffusion barrier, thus distinguishing “two water worlds”.” [L 173-178]

12. In the Methods section describing the theory, why are the words convection, diffusion and connectivity in quotation marks (Lines 256, 257 and 329)?

We meant to draw attention to the important terms convection and diffusion, which are defined at the places pointed out by the referee, appear at many places in the Main Text, and in the Methods. The quotation marks might not be the best way to draw attention to these terms though, so we removed these marks in the revised version of the manuscript. We retained quotation marks around the term “connectivity”, which include the full expression “diffusion connectivity” in the revised version of the manuscript, because we wanted to propose a terminology characterizing the factor $D_p \frac{A_{ij}}{l_{ij}}$. We think that quotation marks are relevant here because this expression is not standard.

13. Line 263, I imagine what is implied is that the D₂O concentration is assumed uniform in the protoplasts. If so, perhaps you should make that explicit.

The referee is correct. This is now clarified as follows in the revised version of the manuscript:

“The diffusivity of D₂O within the protoplast is considered as non-limiting given its high porosity relative to membranes and plasmodesmata, so that D₂O concentration is assumed to freely equilibrate and thus to be uniform within each protoplast.” [L 286-8]

14. In Equation 4 shouldn't the left-hand side of the equality read $\partial(V_i C_i)/\partial t$ to account for possible cell volume changes? Or, are volumes assumed fixed?

The modification suggested by the referee would indeed be required if the volumes of protoplasts or cell walls were to change over time. Such temporal changes of cell volumes continuously occur in the root elongation zone, but could also result from altered cell turgidity due to root dehydration. We did not consider this additional level of complexity in the version of the model published in this manuscript for two main reasons:

- In the conditions of our experiments, the *A. thaliana* roots remained fully hydrated, so that we have no reason to think that cell volumes changes occurred in the differentiation zone, where water is absorbed and then transferred axially in xylem vessels;
- The elongation zone, where the volumes of cells change continuously, is not included in the modelling framework because (i) it does not have functional xylem vessels, so we assumed that it does not substantially affect the composition of water transported axially in mature xylem vessels, and (ii) the elongation zone is a “water sink” due to cell elongation, so we assumed that water absorbed in the elongation zone only contributes to filling elongating cells.

These points were clarified in the revised version of the manuscript as follows:

- V_i is now defined as “the fixed volume of the cell solution allocated to node i ”; [L 329]
- In the general description of the model, we develop the following explanation: “In this study we focus on regions of the root upstream of the observation point which substantially affect xylem water composition. We therefore exclude the elongation zone, which does not have functional xylem vessels and radially absorbs part of the water it needs for cell elongation. For this reason, and because roots stayed fully hydrated during experiments, we assumed that cell volumes represented in the three-dimensional modelling framework did not change over time. Consequently, the root hydraulic anatomy has constant volumes for pieces of compartments, with specific diffusivities at their interfaces. An *ad-hoc* version of the solute advection-diffusion equation accounting for these specificities is solved.” [L 300-7]

15. In Lines 282,286,293,297,299,319,323,326,328, and possibly in other places “I” appears where “i” is meant - probably a MSWord autocorrect error.

We thank the referee for tracking down these autocorrect errors. All of the nine errors were rolled back in the revised version of the manuscript.

16. In Equation 5 the right-hand side requires knowledge of future values of C_t and $\partial C_t / \partial t$. Could the authors explain this conundrum?

The use of the Crank-Nicholson implicit scheme was instrumental in keeping the D_2O advection-diffusion numerical solutions stable, and increasing the time steps to reasonably large values. For that, it solves an equation in which the difference between concentrations at times t and $t+dt$ (C_t and C_{t+dt}) is a function of the temporal derivatives of the concentrations at times t and $t+dt$ ($\partial C_t / \partial t$ and $\partial C_{t+dt} / \partial t$). In order to solve this equation, one must express $\partial C_{t+dt} / \partial t$ as a function of C_{t+dt} . Here this is possible because the water flow field is at steady-state at least from time t to time $t+dt$ (note that D_2O concentration is not at steady-state however), so the matrix \mathbf{M} and vector F_{BC} are known even at time $t+dt$. Once $\partial C_{t+dt} / \partial t$ is expressed as a function of C_{t+dt} , C_{t+dt} can be relocated to the left-hand side (see Eq. 7) without any other unknowns from the time $t+dt$. Eq. 7 can then be solved using the Scipy “spsolve” function.

The explanation was improved in the revised version of the manuscript:

“In order to solve Eq. (5), one must express $\frac{\partial C_{t+dt}}{\partial t}$ as a function of C_{t+dt} , then regroup it to the left-hand side. Here this is possible because the water flow field is at steady-state at least from time t to $t+dt$ (note that D_2O concentration does not need to be at steady-state), so the relation between C and its temporal derivative is known, see Eq. (6).” [351-4]

17. I was a little confused by the similar notation for node value C_i and vector C_t . Could these be distinguished in a more distinctive way?

The referee points at an important syntax clarification requirement. C_i is a scalar while C_t is a vector. Some journals propose to set vectors symbols as bold italics to distinguish them easily from scalars in italics, and matrices in bold. We propose to follow this syntax in the revised manuscript.

After clarifying the symbols for unit types, and before developing equation 1, the revised version of the manuscript now includes the following definition of the syntax for scalars,

vectors and matrices: “For enhanced clarity, scalars are represented by symbols in italics, vectors by symbols in bold italics, and matrices by symbols in bold.” [L 310-1]
The syntax for all vectors symbols (\mathbf{C} , \mathbf{C}_t , \mathbf{C}_{t+dt} , \mathbf{V} , \mathbf{F}_{BC}) was adjusted accordingly in the revised version of the manuscript.

18. In Eq 6, I was also confused by the inconsistency of left and right hand terms of the equation. On the right the two terms represent $N_{tot} \times 1$ vectors, while the right hand side appears to be a $N_{tot} \times N_{tot}$ matrix. This is probably just a poorly expressed way of writing what is meant, but could the authors fix it?

The referee is right. Vectors \mathbf{V} and $\partial \mathbf{C} / \partial t$ on the left-hand side are indeed $N_{tot} \times 1$ sized vectors. We meant to multiply their terms one by one to obtain an $N_{tot} \times 1$ sized vector. However, it could be confused for a “dot product” or “scalar product” with vectors of the wrong size (the first vector would need to be transposed to allow such a product), which would yield a single scalar value whose size would also be a mismatch relative to the right-hand size. In order to avoid this confusion, we rewrote this term in Eq 6 as the dot product “ $\text{diag}(\mathbf{V}) \cdot \partial \mathbf{C} / \partial t$ ” where $\text{diag}(\mathbf{V})$ is an $N_{tot} \times N_{tot}$ diagonal matrix with the vector \mathbf{V} on its main diagonal and zeros elsewhere. Their product is then clearly an $N_{tot} \times 1$ vector. On the right-hand side, the dot product “ $\mathbf{M} \cdot \mathbf{C}$ ” yields a vector with size $N_{tot} \times 1$, which has the same size as \mathbf{F}_{BC} . We believe the use of the diagonal matrix and dot product on the left-hand side combined to the new vector symbol syntax make Eq. 6 clearer in the revised version of the manuscript.

19. In Line 345 et seq, how many stacks are assumed in the model? It is not clear from my reading of the manuscript.

The maximum number of stacks we tested was 280 as the longest distance between the start of the differentiation zone and the laser focal point was about 2.8 cm in one of the replicates. This is now specified in the revised manuscript:

“The cross section is given a third spatial dimension, assuming 10^{-4} m long cells²³ stacked axially (here we tested the model with up to 280 stacks, since the longest distance between the start of the differentiation zone and the laser focal point was about 2.8 cm).” [L 380-3]

However, in the inverse modelling exercise used to estimate the subcellular scale hydraulic parameters, we simulated water flow across 150 stacks or 1.5 cm, which approximated the average root length in L_{pr} measurement. This is now explicitly mentioned in the revised manuscript:

“Note that L_{pr} simulations include the region of the root starting at the point of maturation of protoxylem vessels and consecutive 1.5 cm (150 stacks) shootward.” [L 454-5]

For the estimation of D_2O diffusivities and axial fluxes by inverse modelling, we were able to reduce the number of stacks to 15 because the shapes of xylem D_2O wash-out curves (not their absolute values) were conserved regardless of the total number of stacks (see sensitivity analysis below in WT, in which the distance of the focal point from the barrier varies from 4 to 8 mm, and the length of the differentiated root zone immersed in D_2O varies from 11 to 19.8 mm), see Figure S7 in the reply to referee #1 and in the revised supplementals.

The number of stacks for this inverse modelling scheme is also explicitly mentioned in the revised version of the manuscript:

“Thus, in order to save computing time and resources, the inverse modelling loop is conducted on a shorter virtual root of 0.15 cm (15 stacks) (...)” [L 470-1]

In the revised version of the manuscript, we also clarify that there are 517 nodes per stack after explaining how nodes are defined:

“Apoplastic nodes are located at the center of each apoplastic block and at their junctions (see Fig. 2A, dotted and dashed white circles, respectively) for a total of 517 nodes in the apoplast and symplast within each two-dimensional plane in this study.” [L 290-3]

20. Generally, I kept forgetting what RMS stood for, I kept thinking of root mean square, which is what RMS typically stands for. Perhaps the authors could use something different such as $R_{\mu S}$?

RMS is a field-specific term of art for Raman microspectroscopy, referred to in numerous papers since first appearing in use (*Wu, K. T., et al. "Raman microscopy examination of phase evolution in Bi (Pb)-Sr-Ca-Cu-O superconducting ceramics." Journal of materials research 12.5 (1997): 1195-1204.*). The coincidence with "root mean squared" is indeed unfortunate but we would prefer to adhere to the established terminology.

21. In reading through the description of the theory I was struck by the essential similarity with the model presented by Foster and Miklavcic (Frontiers in Plant Science 2018, 2019, 2021). Although the model here utilizes a root cross section to set up the root (as stacks of cells with this cross section and of 10^{-4} m length) rather than circular arc cylinder segments, as imagined by Foster and Miklavcic, the entire model of discrete equations is identical (naturally), making the calculations effectively the same. So too is the idea of optimizing parameter set against experimental data to deduce physically realistic parameters. The comparison is further enhanced by the fact that Foster and Miklavcic also model an Arabidopsis thaliana root. I would have thought it appropriate to mention this earlier work and comment on likenesses or differences. How do the optimized parameter values compare? Are the differences in specific model features important? Are similar traits found and behaviour found?

We agree with the referee. Similarities between MECHA and the root micro-hydrological model of Foster and Miklavcic were pointed out in a previous publication (Couvreur et al., 2018) but we can update here that both solve advection-diffusion equations, each with their own specificities. Based on the set of equations published in Foster and Miklavcic (2019)²⁴, my understanding is that water flow and solute advection-diffusion equations, as well as boundary conditions, are equivalent to ours. Paths also include cell walls, membranes and plasmodesmata in both models, which have been used to offer new insights into water and solute transport in *A. thaliana*.

Besides their similarities, we think the specificities of the two models are strengths that make them complementary. A specificity of MECHA mentioned by the referee is its three spatial dimensions, which allows capturing complex anatomical features that are not radially symmetrical (e.g. aerenchyma, xylem and phloem tissues). MECHA also has its Casparian strip in the radial walls of the endodermis, which is important in this study as we put a central focus on the localisation of apoplastic barriers. Specificities of the model of Foster and Miklavcic are many. Here we list a few and apologize for any that we missed: it captures the transport of multiple solutes simultaneously; these solutes can be ions with associated active or passive membrane transporters across the plasmalemma and vacuolar membrane; It also accounts for an impressive range of other processes such as

H⁺ buffering, electric potential, the binding of cations to fixed anions, etc. In terms of solver, MECHA uses “spsolve” to obtain the numerical solution of the implicit “Crank-Nicholson” scheme from sparse matrices and vectors. We found that Foster and Miklavcic use the *ode15.m* function to solve the ordinary differential equations, but it is not clear to us if the scheme is explicit. If that is the case, and if their system requires particularly small time steps to allow convergence, implementing an implicit scheme like in MECHA might offer the opportunity to use larger time steps and work on larger roots, thanks to the reduced computational time. It is not clear to us if implementing such a scheme is possible in their model though.

We thank the referee for suggesting the interesting comparison of our diffusion and hydraulic constants to the ones found by Foster and Miklavcic²⁴. In their study, diffusion coefficients were estimated for small ions, whose size is of the same order of magnitude as water molecules, and found similar diffusion coefficients values to ours. Comparing the sub-cellular hydraulic parameters has been difficult for us as the “radial apoplastic water permeability” parameter value is not explicit in their table S2. It is divided by d_a , whose value we did not find in the supplemental table of parameter values. It is also not clear to us how to compare the “symplastic water permeability” to our hydraulic conductance per plasmodesmata. Therefore, we made a comparison with values found by inverse modelling using MECHA and L_{pr} data in maize, and briefly discuss differences.

In the revised version of the manuscript, we make the following statement when presenting the model and the optimized parameters:

“For more elaborate simulations of the transport of major ions and associated processes in *A. thaliana* roots, an axisymmetric radial-longitudinal model has been developed by Foster and Miklavcic^{2,24}.” [L 282-4]

“Diffusion coefficients with similar values to those we found have been reported for small ionic compounds such as Na⁺ and K⁺ in *A. thaliana* cell walls², which we interpret as a positive sign in view of the large uncertainties associated with the estimation of diffusion coefficients in plant tissues²⁵. Apart from the hydraulic conductivity of membranes, which matches the range found in maize²⁶, the sub-cellular hydraulic conductivities of cell walls and plasmodesmata are smaller than those found in maize with the same modelling framework²⁷. Reduced diffusion coefficients and hydraulic conductivities in *A. thaliana* porous media are likely due to lower porosity, constrictivity, and higher tortuosity²⁸.” [L 431-8].

22. Figures 1 and 2 are not so easy to read. Indeed in Figure 1 (D-H) it is difficult to read the figure axes (titles and values). The same applies to Figure 2 (F-G). Also, Figure 1 (A) is rather difficult to make out, despite the labelling.

We thank the reviewer for pointing out these issues. These figures have been redrafted to improve legibility and signpost the reader, as referred to above.

In Fig 1, changes include removing former panel B to directly connect panel A to former panel C, and optimizing the layout and sizing of text, arrows, and lines.

In Fig 2, character sizes were increased in panels A, B, C, D, E, F, and G. Arrow sizes were increased in panel C. The term “convection” was renamed “advection” in panel D.

While I understand the origin of Figure 1 (C), I am not sure I understand what I am looking at. Is it a longitudinal section of the root, or is it just a enlarged view of the root exterior surface.

This bright field micrograph has been reoriented and signposted at a larger size to guide the reader. The figure is now labelled "bright field" and the caption similarly reflects this, along with pointing out the visible xylem that confirm the centrality of the plane of focus. Our objective lens has a rather high NA which gives quite a tight working volume which, despite relatively poor bright field contrast, is capable of delineating anatomy to some degree. Our guide to location chosen however was provided by RMS signal, with bright field simply a useful orientation guide.

I would appreciate a revision of these figures to make them more legible and easier to comprehend.

The figures have been duly revised and, we believe, rather improved based on this useful feedback – thank you. The new version of Fig 1 is the following:

The new version of Fig 2 is the following:

Recommendation: I recommend acceptance subject to minor revision.

References

- 1 Pascut, F. C. *et al.* Non-invasive label-free monitoring the cardiac differentiation of human embryonic stem cells in-vitro by Raman spectroscopy. *Biochimica et Biophysica Acta (BBA) - General Subjects* **1830**, 3517-3524, doi:<https://doi.org/10.1016/j.bbagen.2013.01.030> (2013).
- 2 Foster, K. J. & Miklavcic, S. J. A Comprehensive Biophysical Model of Ion and Water Transport in Plant Roots. I. Clarifying the Roles of Endodermal Barriers in the Salt Stress Response. *Frontiers in Plant Science* **8**, doi:10.3389/fpls.2017.01326 (2017).
- 3 Foster, K. J. & Miklavcic, S. J. A Comprehensive Biophysical Model of Ion and Water Transport in Plant Roots. III. Quantifying the Energy Costs of Ion Transport in Salt-Stressed Roots of Arabidopsis. *Frontiers in Plant Science* **11**, doi:10.3389/fpls.2020.00865 (2020).
- 4 Knipfer, T. & Steudle, E. Root hydraulic conductivity measured by pressure clamp is substantially affected by internal unstirred layers. *J. Exp. Bot.* **59**, 2071-2084, doi:10.1093/jxb/ern064 (2008).
- 5 Molz, F. Advection, Dispersion, and Confusion. *Groundwater* **53**, 348-353, doi:<https://doi.org/10.1111/gwat.12338> (2015).
- 6 Malcolm, D. T. K., Hunter, P. J., Donaldson, P., Kistler, J. & Mathias, R. T. Modelling the circulation in the mammalian lens. *IFAC Proceedings Volumes* **36**, 105-107, doi:[https://doi.org/10.1016/S1474-6670\(17\)33482-1](https://doi.org/10.1016/S1474-6670(17)33482-1) (2003).
- 7 Boursiac, Y. *et al.* Early Effects of Salinity on Water Transport in Arabidopsis Roots. Molecular and Cellular Features of Aquaporin Expression. *Plant Physiology* **139**, 790-805, doi:10.1104/pp.105.065029 (2005).
- 8 Tournaire-Roux, C. *et al.* Cytosolic pH regulates root water transport during anoxic stress through gating of aquaporins. *Nature* **425**, 393-397 (2003).
- 9 Wang, P. *et al.* Surveillance of cell wall diffusion barrier integrity modulates water and solute transport in plants. *Scientific Reports* **9**, 4227, doi:10.1038/s41598-019-40588-5 (2019).
- 10 Reyt, G. *et al.* Two chemically distinct root lignin barriers control solute and water balance. *Nature Communications* **12**, 2320, doi:10.1038/s41467-021-22550-0 (2021).
- 11 Naseer, S. *et al.* Casparian strip diffusion barrier in Arabidopsis is made of a lignin polymer without suberin. *Proceedings of the National Academy of Sciences* **109**, 10101-10106, doi:10.1073/pnas.1205726109 (2012).
- 12 Barberon, M. *et al.* Adaptation of Root Function by Nutrient-Induced Plasticity of Endodermal Differentiation. *Cell* **164**, 447-459 (2016).
- 13 Pérez-Pérez, J. G. & Dodd, I. C. Sap fluxes from different parts of the rootzone modulate xylem ABA concentration during partial rootzone drying and re-wetting. *Journal of experimental botany* **66**, 2315-2324, doi:10.1093/jxb/erv029 (2015).
- 14 Robert, H. S. & Friml, J. Auxin and other signals on the move in plants. *Nature Chemical Biology* **5**, 325-332, doi:10.1038/nchembio.170 (2009).

- 15 Burley, J. W. A., Festus, I. O. N., Leister, G. L. & Popham, R. A. The Relationship of Xylem Maturation to the Absorption and Translocation of P³². *Amer. J. Bot.* **57**, 504-511, doi:10.2307/2441047 (1970).
- 16 Rygol, J., Pritchard, J., Zhu, J. J., Tomos, A. D. & Zimmermann, U. Transpiration Induces Radial Turgor Pressure Gradients in Wheat and Maize Roots. *Plant Physiol.* **103**, 493-500 (1993).
- 17 Steudle, E. & Jeschke, W. D. Water transport in barley roots : Measurements of root pressure and hydraulic conductivity of roots in parallel with turgor and hydraulic conductivity of root cells. *Planta* **158**, 237-248 (1983).
- 18 Song, Y., Kirkham, M. B., Ham, J. M. & Kluitenberg, G. J. Root-zone hydraulic lift evaluated with the dual-probe heat-pulse technique. *Aust. J. Soil Res.* **38**, 927-935 (2000).
- 19 Deng, J., Xiong, T., Wang, H., Zheng, A. & Wang, Y. Effects of Cellulose, Hemicellulose, and Lignin on the Structure and Morphology of Porous Carbons. *ACS Sustain. Chem. Eng.* **4**, 3750-3756, doi:10.1021/acssuschemeng.6b00388 (2016).
- 20 Dawson, T. E. & Ehleringer, J. R. Streamside trees that do not use stream water. *Nature* **350**, 335-337, doi:10.1038/350335a0 (1991).
- 21 Lewis, B. A. & Engelman, D. M. Lipid bilayer thickness varies linearly with acyl chain length in fluid phosphatidylcholine vesicles. *Journal of Molecular Biology* **166**, 211-217, doi:https://doi.org/10.1016/S0022-2836(83)80007-2 (1983).
- 22 Ehlers, K. & Kollmann, R. Primary and secondary plasmodesmata: structure, origin, and functioning. *Protoplasma* **216**, 1-30, doi:10.1007/bf02680127 (2001).
- 23 De Cnodder, T., Vissenberg, K., Van Der Straeten, D. & Verbelen, J.-P. Regulation of cell length in the Arabidopsis thaliana root by the ethylene precursor 1-aminocyclopropane- 1-carboxylic acid: a matter of apoplastic reactions. *New Phytologist* **168**, 541-550, doi:10.1111/j.1469-8137.2005.01540.x (2005).
- 24 Foster, K. J. & Miklavcic, S. J. A Comprehensive Biophysical Model of Ion and Water Transport in Plant Roots. II. Clarifying the Roles of SOS1 in the Salt-Stress Response in Arabidopsis. *Frontiers in Plant Science* **10**, doi:10.3389/fpls.2019.01121 (2019).
- 25 Kramer, E. M., Frazer, N. L. & Baskin, T. I. Measurement of diffusion within the cell wall in living roots of Arabidopsis thaliana. *Journal of Experimental Botany* **58**, 3005-3015, doi:10.1093/jxb/erm155 (2007).
- 26 Ehler, C., Maurel, C., Tardieu, F. & Simonneau, T. Aquaporin-Mediated Reduction in Maize Root Hydraulic Conductivity Impacts Cell Turgor and Leaf Elongation Even without Changing Transpiration. *Plant Physiol.* **150**, 1093-1104 (2009).
- 27 Couvreur, V. *et al.* Going with the Flow: Multiscale Insights into the Composite Nature of Water Transport in Roots. *Plant Physiology* **178**, 1689-1703, doi:10.1104/pp.18.01006 (2018).
- 28 Grathwohl, P. Diffusion in natural porous media : contaminant transport, sorption/desorption and dissolution kinetics. (1998).

REVIEWERS' COMMENTS

Reviewer #1 (Remarks to the Author):

The authors addressed my concerns. I do not have other comments.

Reviewer #2 (Remarks to the Author):

I would like to thank the authors for thoughtfully addressing my concerns. In my view the manuscript is now acceptable and will make a lasting impact on the field.

Reviewer #3 (Remarks to the Author):

I am happy with the changes made to the manuscript. I think it reads much better. Figures 1 and 2 are also much clearer. Thank you for taking the time to make these changes. I am happy to recommend publication.

In this document, comments by reviewers appear in black and **replies by the authors in blue.**

Reviewer #1 (Remarks to the Author):

The authors addressed my concerns. I do not have other comments.

We thank Reviewer #1 for taking the time to go through our replies and for confirming that there are no remaining concerns in the revised version of the manuscript.

Reviewer #2 (Remarks to the Author):

I would like to thank the authors for thoughtfully addressing my concerns. In my view the manuscript is now acceptable and will make a lasting impact on the field.

We would like to reiterate our gratitude to Reviewer #2 for the extensive and insightful comments, which greatly helped improving the manuscript. We are particularly heartened by the reviewer's view that the manuscript will make a lasting impact on the field, and will keep working to produce impactful research in the field with our novel combined RMS – modelling approach.

Reviewer #3 (Remarks to the Author):

I am happy with the changes made to the manuscript. I think it reads much better. Figures 1 and 2 are also much clearer. Thank you for taking the time to make these changes. I am happy to recommend publication.

We thank Reviewer #3 for the thorough evaluation of the manuscript, which greatly helped to make it clearer and more accessible to the broad readership of Nature Communications. In particular, we very much appreciated that the reviewer offered comments on the mathematical developments.